# Single spikes drive sequential propagation and routing of activity in a cortical network

Juan Luis Riquelme[1,2], Mike Hemberger[1], Gilles Laurent[1], Julijana Gjorgjieva[1,2]*

[1]Max Planck Institute for Brain Research, Frankfurt am Main, Germany; [2]School of Life Sciences, Technical University of Munich, Freising, Germany

**Abstract** Single spikes can trigger repeatable firing sequences in cortical networks. The mechanisms that support reliable propagation of activity from such small events and their functional consequences remain unclear. By constraining a recurrent network model with experimental statistics from turtle cortex, we generate reliable and temporally precise sequences from single spike triggers. We find that rare strong connections support sequence propagation, while dense weak connections modulate propagation reliability. We identify sections of sequences corresponding to divergent branches of strongly connected neurons which can be selectively gated. Applying external inputs to specific neurons in the sparse backbone of strong connections can effectively control propagation and route activity within the network. Finally, we demonstrate that concurrent sequences interact reliably, generating a highly combinatorial space of sequence activations. Our results reveal the impact of individual spikes in cortical circuits, detailing how repeatable sequences of activity can be triggered, sustained, and controlled during cortical computations.

## Editor's evaluation

This is an important study of the role of spike timing in the turtle cortex. The authors provide compelling evidence that single spikes evoke motifs via strong connections, and that those motifs can be reliably routed by weaker connections. The work is careful and clear and makes intuitive predictions about how motifs are generated. It will be especially interesting to determine to what extent the results apply to the mammalian cortex.

*For correspondence:
gjorgjieva@tum.de

## Introduction

Experimental and modeling studies have proposed that cortical circuits rely on firing rates to convey information reliably in the presence of irregular activity and intrinsic sources of noise (*London et al., 2010*; *Renart et al., 2010*; *Shadlen and Newsome, 1994*). This suggests that individual spikes are effectively superfluous for computation. On the other hand, even a single spike can noticeably increase network firing rates in rat barrel cortex (*London et al., 2010*) and trigger reliable sequences of spikes in human and turtle cortex (*Hemberger et al., 2019*; *Molnár et al., 2008*). Indeed, repeatable and temporally precise patterns of action potentials in neural circuits have long suggested that the precise timing of spikes may play an important role (*Bair and Koch, 1996*; *Bolding and Franks, 2017*; *Fellous et al., 2004*; *Hahnloser et al., 2002*; *Kumar et al., 2010*; *Wehr and Laurent, 1996*; *Wehr and Zador, 2003*). How relevant, therefore, are single spikes for cortical function? Because spikes are the main form of neuronal communication, understanding how networks respond to single spikes is crucial to define the building blocks of cortical computation (*Brette, 2015*).

**eLife digest** Neurons in the brain form thousands of connections, or synapses, with one another, allowing signals to pass from one cell to the next. To activate a neuron, a high enough activating signal or 'action potential' must be reached. However, the accepted view of signal transmission assumes that the great majority of synapses are too weak to activate neurons. This means that often simultaneous inputs from many neurons are required to trigger a single neuron's activation. However, such coordination is likely unreliable as neurons can react differently to the same stimulus depending on the circumstances. An alternative way of transmitting signals has been reported in turtle brains, where impulses from a single neuron can trigger activity across a network of connections. Furthermore, these responses are reliably repeatable, activating the same neurons in the same order.

Riquelme et al. set out to understand the mechanism that underlies this type of neuron activation using a mathematical model based on data from the turtle brain. These data showed that the neural network in the turtle's brain had many weak synapses but also a few, rare, strong synapses. Simulating this neural network showed that those rare, strong synapses promote the signal's reliability by providing a consistent route for the signal to travel through the network. The numerous weak synapses, on the other hand, have a regulatory role in providing flexibility to how the activation spreads. This combination of strong and weak connections produces a system that can reliably promote or stop the signal flow depending on the context.

Riquelme et al.'s work describes a potential mechanism for how signals might travel reliably through neural networks in the brain, based on data from turtles. Experimental work will need to address whether strong connections play a similar role in other animal species, including humans. In the future, these results may be used as the basis to design new systems for artificial intelligence, building on the success of neural networks.

The influence of single neurons has been documented in rat and mouse cortex, where single-cell stimulation has meso- and macroscopic effects on network activity, brain state, and even behavior (*Brecht et al., 2004*; *Doron et al., 2014*; *Houweling and Brecht, 2008*; *Kwan and Dan, 2012*; *Wolfe et al., 2010*). Similarly, recent experiments in turtle cortex have shown that one to three spikes of a single neuron can trigger activity in the surrounding network (*Hemberger et al., 2019*). Only containing three layers, turtle visual cortex is architectonically similar to mammalian olfactory cortex or hippocampus and evolutionarily linked to the six-layered mammalian neocortex (*Fournier et al., 2015*; *Tosches et al., 2018*). Moreover, it lends itself to long ex vivo experiments where local connectivity is intact. Recent experiments found that the activity triggered by electrically evoked single spikes in the turtle cortex is reliable in three ways: responses are repeatable across trials, the responses involve the same neurons, and their activations respect the same temporal order (*Hemberger et al., 2019*). Repeatable sequences of spikes have been reported across various systems in vivo, such as in replay or pre-play in rat hippocampus (*Buzsáki and Tingley, 2018*; *Diba and Buzsáki, 2007*; *Dragoi and Tonegawa, 2011*), rat auditory and somatosensory cortex (*Luczak et al., 2015*; *Luczak and Maclean, 2012*), mouse visual and auditory cortex (*Carrillo-Reid et al., 2015*; *Dechery and MacLean, 2017*), and human middle temporal lobe (*Vaz et al., 2020*). Although often linked to behavioral or sensory cues, the network mechanisms that underlie such sequences are unknown. Even in the ex vivo turtle cortex, the electrical distance between the MEA and pyramidal cell layer has limited the observation of sequence propagation within the excitatory neuron population (*Fournier et al., 2018*; *Shein-Idelson et al., 2017*). How sequences propagate, even under irregular and seemingly noisy activity, remains a puzzle, yet it may be key to understanding the computational role of sequences and cortical computations more generally (*Buzsáki and Tingley, 2018*).

Several candidate theoretical frameworks based on structured connectivity could explain the reliable propagation of activity during sequences. One example is synfire chains, where divergent–convergent connectivity might connect groups of neurons that fire synchronously (*Abeles, 1991*; *Diesmann et al., 1999*; *Kumar et al., 2010*), or polychronous chains, where transmission delays require neurons to fire in precise time-locked patterns (*Izhikevich, 2006*). However, experimental evidence for these specialized connectivity structures is limited (*Egger et al., 2020*; *Ikegaya et al., 2004*; *Long et al., 2010*). Alternatively, structured connectivity leading to sequences has been proposed to arise via the

training of connections (*Fiete et al., 2010*; *Maes et al., 2020*; *Rajan et al., 2016*). Finally, models based on the turtle cortex, in particular, have investigated network-wide propagation of waves or the statistical properties of population activity in the form of neuronal avalanches (*Nenadic et al., 2003*; *Shew et al., 2015*). Overall, these models of cortical propagation focus primarily on coordinated population activity and have not investigated how single-neuron activation can trigger reliable sequences.

To mechanistically investigate sequence generation from single spikes, we built and explored a model network constrained by previously reported experimental measurements of the turtle cortex (*Hemberger et al., 2019*). Our model readily generates reliable sequences in response to single-neuron activations without any fine-tuning. Analyzing the properties of sequences as a function of model parameters, we found that few strong connections support sequence propagation while many weak connections provide sequence flexibility. We examined how sparse contextual input modulates sequences, modifying the paths of propagation of network activity. Our results suggest that single spikes can reliably trigger and modify sequences and thus provide reliable yet flexible routing of information within cortical networks.

## Results

### Model of the turtle visual cortex with experimentally defined constraints

To explore the network mechanisms that might lead to the reliable activity propagation from single spikes, we built a recurrent network model with single-cell properties and connectivity constrained to previously obtained experimental measurements from the turtle visual cortex (*Hemberger et al., 2019*). The network model is composed of 100,000 neurons (93% excitatory and 7% inhibitory) equivalent, by neuronal density, to a 2 × 2 mm slab of turtle visual cortex (*Figure 1A*).

We modeled neurons as adaptive exponential integrate-and-fire units based on experimental evidence that excitatory (pyramidal) neurons show adaptive spiking (*Hemberger et al., 2019*). We used previously found adaptation current parameters to capture the median of the experimental distribution of the Adaptation index (0.3) in the turtle cortex (*Brette and Gerstner, 2005*). We fitted membrane capacitance and leak conductance parameters to match membrane potential from previous experimental recordings under current-clamp and used the median values for our model (*Figure 1B*).

As suggested by axonal arbor reconstructions of turtle cortex neurons (*Hemberger et al., 2019*), we connected model neurons with probabilities that decay spatially. We fitted a Gaussian profile for the spatial decay to connection probability estimates that had been obtained from paired whole-cell patch-clamp recordings at multiple distances (*Hemberger et al., 2019*; *Figure 1C*). Because our estimates of connection probabilities (*Figure 1A*) are derived from 'paired-patch' recordings of nearby neurons, we scaled the decay profile to match population-specific probabilities in any given disc of 200 μm radius.

Direct and indirect evidence from paired-patch experiments in turtle cortex indicates the presence of rare but powerful excitatory synapses, resulting in a long-tailed distribution of excitatory postsynaptic potential (EPSP) amplitudes (*Hemberger et al., 2019*). We thus generated excitatory synaptic conductances in our model from a lognormal distribution fitted to the experimental data (*Figure 1D*, see Methods, Neuron and synapse models). We truncated the distribution at the conductance value corresponding to the largest recorded EPSP amplitude in our paired-patch experiments (21 mV). Based on these experimental constraints, each model excitatory neuron connects to other neurons with a majority of weak synapses and about two connections sufficiently strong to bring the postsynaptic neuron from rest to near firing threshold (*Figure 1D, inset, E*). The actual efficacy of these rare but powerful connections depends on the conductance state of their postsynaptic neurons and, thus, on ongoing network activity and the state of adaptation currents. To verify that our lognormal fit is not strongly biased by the experimental dataset, we bootstrapped our estimate for the probability of a neuron having at least one strong connection under our connectivity assumptions (*Figure 1F*). We found a heavily skewed distribution, with 39% of bootstrapped fits showing a probability even higher than our model.

We generated inhibitory connections from a scaled-up copy of the distribution of excitatory conductances, resulting in a skewed distribution, as observed in other systems (*Iascone et al.,*

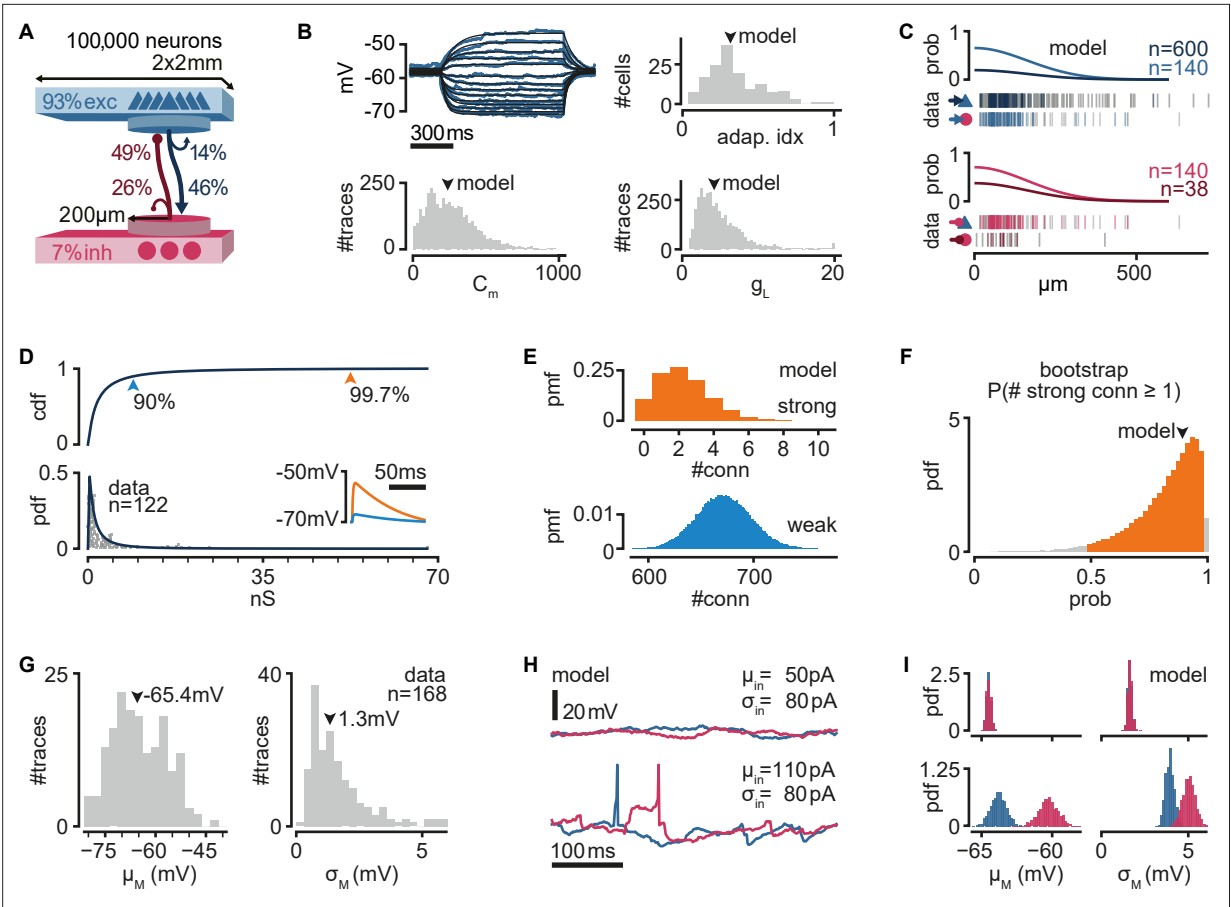

**Figure 1.** A biologically inspired model of turtle visual cortex. (**A**) Schematic of the network with the size, proportion of E and I populations, and connection probabilities within a disc of 200 μm radius (see **C**). Mean number of outgoing connections per cell: E-to-E: 750, I-to-I: 110, E-to-I: 190, I-to-E: 2690. Blue triangles: excitatory neurons. Red circles: inhibitory neurons. (**B**) Top left: example of fit (black) of membrane potential responses to current injection in a recorded neuron (blue). Bottom left and right: distributions of fitted membrane capacitance ($C_m$) and leak conductance ($g_L$) ($n = 3886$ traces). Top right: distribution of adaptive indices for 145 recorded neurons. Arrowheads indicate parameters of model neurons (median). (**C**) Gaussian profiles of distance-dependent connection probabilities for different connection types (top: exc; bottom: inh). The profiles were fitted from the fraction of pairs of patched neurons that showed a connection from all tested pairs. Vertical bars below (data) indicate connected (colored) or disconnected (gray) pairs of neurons for different connection types. (**D**) Lognormal fit to peak excitatory synaptic conductances. Top: cumulative distribution function (cdf). Bottom: probability density function (pdf). Conductances were estimated from excitatory postsynaptic potential (EPSP) amplitudes experimentally obtained from recorded pairs of connected neurons (gray dots). Inset: example of modeled EPSPs for different synaptic weights (top arrowheads) in an isolated postsynaptic neuron at resting potential. (**E**) Probability mass function (pmf) of number of strong (top 0.3%) or weak (bottom 90%) excitatory-to-excitatory connections per model neuron. (**F**) Bootstrap estimate of the probability of an excitatory neuron having at least one strong excitatory-to-excitatory connection. Colored area: 95% confidence interval. Arrowhead: fit with the original dataset in (**D**). (**G**) Experimentally measured mean and standard deviation of the membrane potential of 168 patched neurons in ex vivo preparations. Arrowheads indicate medians. (**H**) Membrane potential of two model neurons (blue: exc; red: inh) under different white noise current parameters. Note the presence of action potentials and EPSPs under a high mean current ($\mu_{in}$). (**I**) Distributions of membrane potential mean and standard deviation for model neurons (blue: exc; red: inh) under the white noise current parameters in (**H**).

2020; *Kajiwara et al., 2020*; *Rubinski and Ziv, 2015*). We assumed that strengths for excitatory and inhibitory connections are independent of distance. For each connection, we added a random delay between 0.5 and 2 ms to account for the short and reliable latency between presynaptic action potential and monosynaptic EPSPs in our paired-patch experiments. Besides the spatial connectivity profiles and the lognormal distribution of synaptic strengths for excitatory and inhibitory connections, we otherwise assumed random connectivity.

Neurons recorded in ex vivo conditions generate spontaneous action potentials once every few seconds and exhibit a noisy resting membrane potential (*Hemberger et al., 2019*; *Figure 1G*). We modeled the source of this spontaneous activity as a white noise current sampled independently but

with the same statistics for every simulated neuron. In every simulation, we sampled current parameters uniformly between 50 and 110 pA mean ($\mu_{in}$) and 0–110 pA standard deviation ($\sigma_{in}$). These currents produce various levels of membrane potential fluctuations and spontaneous firing, similar to those observed experimentally (*Figure 1H*). We observed that current mean has a more substantial effect than current variance on the membrane potential variance of model neurons (*Figure 1I*). This results from self-sustained firing in the network at sufficiently depolarizing currents (note spikes in *Figure 1H*, bottom left).

In summary, we built a model where we constrained single-neuron properties and connection strength distributions by biological data but otherwise assumed random connectivity. The model, which reproduces experimental cortical data, displays highly heterogeneous connectivity and provides a testbed to investigate the mechanistic underpinnings of sequence generation and propagation observed in the biological networks.

## Single spikes trigger reliable sequential activity in a biologically constrained network model

Next, we examined if our biologically constrained model produces reliable sequential neuron activations triggered by a single spike in a randomly chosen pyramidal neuron. We first generated 300 random networks and randomly selected an excitatory neuron in each one (called trigger neuron). For each network, we generated 20 simulations under different levels of spontaneous activity, yielding a total of 6000 simulations (*Figure 2—figure supplement 1A*). In each simulation, we caused the trigger neuron to fire 100 action potentials at 400-ms intervals (each action potential defining a trial) and measured, in all the other neurons, the resulting firing rate modulation (ΔFR, *Figure 2A*). To identify reliably activated model neurons, we performed a statistical test on the distribution of ΔFR, assuming a Poisson process of constant firing rate as the null distribution (*Figure 2—figure supplement 1E*). We call 'followers' those model neurons displaying statistically high ΔFR (p = $10^{-7}$, *Figure 2B*, see Methods, Definition of followers). We found followers in 94.6% of our 6000 simulations.

When ordering the followers in the model network by activation delay from the trigger neuron spike, we observed reliable spike sequences as seen in experiments (*Figure 2C*, *Figure 2—figure supplement 1B*). We used an entropy-based measure to quantify the variability of follower identity per rank in a sequence, and observed similar results as in the experiments, with follower ordering being most predictable in the first four ranks of the sequence (see Methods, Entropy of follower rank, *Figure 2D*). As observed in experiments, we found that sequences in our model evolve over space, spreading away from the trigger neuron (*Figure 2E*, left). Furthermore, around 18% of all followers in the model fall outside the frame used in the experimental recordings (defined by a 1.3 × 1.3 mm MEA, stippled line, *Figure 2E*, right), suggesting a possible underestimation of the number of followers and sequence length in the original experiments.

While the electrical distance between the MEA and the pyramidal cell layer limited the experimental observation of excitatory followers in the turtle cortex, our model predicts that their number far exceeds that of inhibitory followers (*Figure 2F*). We tested the effect of spontaneous activity in the network on the number of followers. Our model generates almost no followers when spontaneous firing rates are close to zero. The number of model followers increases rapidly with increasing spontaneous firing rates, peaking at low firing rates (0.007 spk/s mean firing rate for the top 1% simulations by follower count), and slowly decreases as spontaneous firing rates increase further. We classified our simulations into *low* and *high activity* using spontaneous mean firing rate ranges observed in experiments (*Hemberger et al., 2019*): [0, 0.05] spk/s ex vivo, 41% of simulations; [0.02, 0.09] spk/s in vivo, 33% of simulations ([25th, 75th] percentiles).

We found sequences in both groups but with notable differences in the behavior of excitatory and inhibitory model followers. The probability of a low-activity simulation displaying at least 1 or 10 inhibitory followers (74% and 19%, respectively) shows an excellent agreement with experimental data (77% and 20%, respectively, *Figure 2G*). Interestingly, we observed that the probability of producing inhibitory followers drops in high-activity simulations (4% prob. of ≥10 inhibitory followers). Excitatory neurons show the opposite trend: the probability of having at least 10 excitatory followers is higher in high-activity (69%) than in low-activity simulations (50%).

We saw that the sequences of activations of excitatory followers often last more than 150 ms and reach the spatial boundaries of our model circuit (2 × 2 mm, 100k neurons) in both low- and high-activity

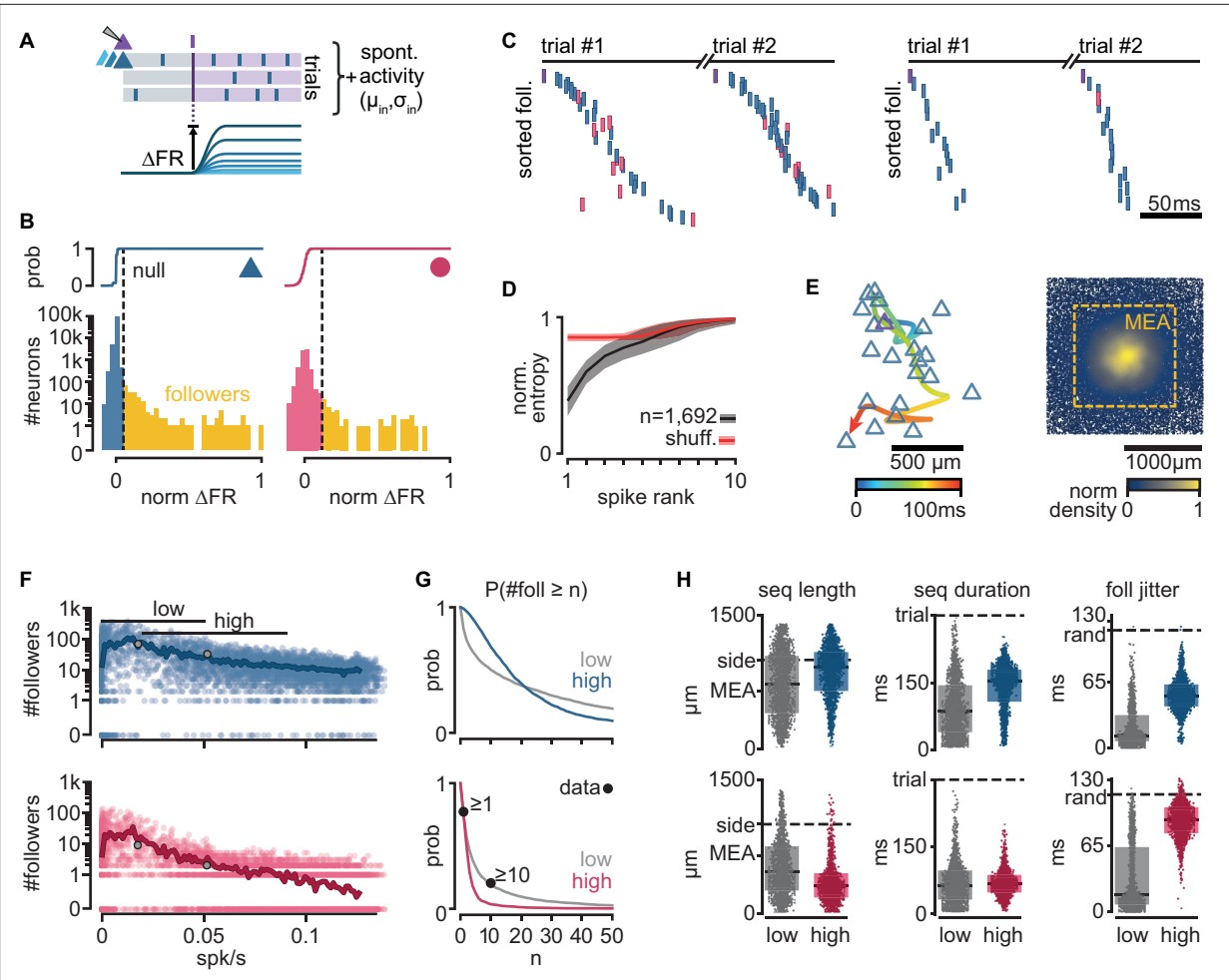

**Figure 2.** Repeatable sequential activation of follower neurons in the model network. (**A**) Schematic of the stimulation protocol in the model. One neuron (top) was driven to produce one action potential, and all other neurons in the network were tested for resulting changes in their firing rate. (**B**) Distribution of single-cell firing rate modulation (ΔFR) for one representative simulation. Top: cumulative mass function of the null distribution. Dashed: threshold for follower classification (p = 10⁻⁷). Bottom: ΔFR for followers (yellow) and other neurons in the network (blue: exc, red: inh). ΔFR is normalized to the modulation of a perfect follower (exactly one spike per trial). (**C**) Left: sequence of spikes from followers in two consecutive trials in an example simulation. Y-axis: followers sorted by median spike time across all trials. Same sorting in both trials. Spikes from non-followers are not shown. Right: same for a different simulation with a higher mean firing rate (left: 0.017 spk/s, right: 0.051 spk/s). See *Figure 2—figure supplement 1C* for full rasters. (**D**) Normalized spike rank entropy of sequences with at least 10 followers (black: mean; gray: std), compared to shuffled sequences (red). (**E**) Left: spatial evolution of the center-of-mass of follower activations during the first 100 ms of the first sequence in (**C**). Trigger neuron in purple outline. Exc followers in blue outlines. Right: spatial location of excitatory followers pooled from all simulations, colored by local density (n = 117,427). All simulations are aligned so that the trigger neuron is in the center. Stippled square: size of the MEA used in experiments. (**F**) Number of followers detected for each simulation as a function of the mean level of activity in the network. Blue: exc; red: inh; thick line: moving average; gray dots: sequences in (**C**). (**G**) Probability of generating a minimum number of followers for excitatory and inhibitory populations in high- and low-activity simulations. Dots: experimental ex vivo estimates. (**H**) Statistics of excitatory- or inhibitory-follower activations by activity level. Left: distance from trigger neuron to farthest detected follower in each simulation (side: half-width of the model network; MEA: half-width of MEA used in experiments). Middle: delay to median spike time of the last activated follower in each simulation (trial: maximum detectable duration under protocol in **A**). Right: standard deviation of follower spike times, averaged over all followers in each simulation (rand: expected standard deviation of randomly distributed spike times). Boxes: median and [25th, 75th] percentiles.

The online version of this article includes the following figure supplement(s) for figure 2:

**Figure supplement 1.** Network firing rates.

simulations (*Figure 2H*, left). Inhibitory followers in the model network, however, often fire early in the sequence and are spatially closer to the trigger neuron than excitatory ones (*Figure 2H*, middle and right). These differences are more pronounced in high- than low-activity simulations (*Figure 2H*, gray and colored). Importantly, inhibitory followers display near-random levels of temporal jitter in

high-activity regimes (*Figure 2H*, foll. jitter, standard deviation of spike delays). Consequently, we predict that followers should be detectable in experiments in vivo in the turtle cortex but mainly within the pyramidal layer.

In summary, our biologically constrained model produces repeatable firing sequences across groups of neurons in response to single spikes in randomly chosen excitatory neurons with properties very similar to those observed in ex vivo experiments (low firing rates) that constrained the model network. Our simulations further provide an experimentally testable prediction: that sequences may occur under in vivo levels of spontaneous firing rate in the turtle cortex. In these conditions of higher spontaneous activity, the activation sequences are mainly composed of excitatory followers, whereas inhibitory followers produce less reliable and temporally jittered responses.

## Strong connections provide reliability of activity propagation while weak connections modulate it

To better understand the mechanisms behind follower activation, we examined how spikes propagate through our model networks. We found that single strong sparse connections drive reliable excitatory responses, while convergent weak connections control the level of spontaneous network activity and drive inhibition.

In a random subset of our 6000 simulations (*n* = 900), we searched for instances when an neuron spike successfully traversed a connection, that is, when the postsynaptic neuron fired within 100 ms of the presynaptic one. We call this successful activation of a connection a 'spike transfer' from the pre- to the postsynaptic neuron. We thus combined spikes with recurrent connectivity to produce a directed acyclic graph of all spike transfers for each simulation (*Figure 3A*). Note that we studied spike transfers among all connected neurons in the network, allowing us to characterize the connections that are most frequently traversed. Most excitatory-to-excitatory spike transfers in low-activity simulations show a delay from pre- to postsynaptic spike of 6–8 ms (*Figure 3B*), matching delays measured in turtle cortex (*Hemberger et al., 2019*). Interestingly, excitatory-to-inhibitory spike transfers display consistently shorter delays than their excitatory-to-excitatory counterparts, even at higher firing rates (*Figure 3B*, inset), possibly reflecting the more depolarized state of inhibitory neurons (*Figure 1I*).

To understand the connectivity structure underlying these spike transfers, we examined connectivity motifs in the graph (*Figure 3C, D*). The possible number of motifs increases with the number of neurons involved. Due to this combinatorial nature, we extracted only low-order motifs with depths of up to two spike transfers and involving ≤4 spikes. In low-activity simulations, we found that excitatory spikes are rarely triggered by convergence or fan motifs (*Figure 3C*, top, conv.), but they are rather the result of one-to-one spike transfers (*Figure 3C*, top, single). By contrast, we saw that spikes from inhibitory neurons result from more spike transfers, with a prevalence of convergence motifs (*Figure 3C*, bottom, conv.). Although spike transfers by convergence are more common in high-activity simulations for both excitatory and inhibitory populations (*Figure 3D*), the increase is greater for motifs leading to inhibitory spikes. Indeed, inhibitory spikes in high-activity simulations rarely involve single spike transfers (*Figure 3D*, bottom, single). This analysis reveals that different excitatory and inhibitory connectivity motifs underlie activity propagation in the networks.

Finally, we extracted the synaptic strengths involved in these spike transfers (*Figure 3E*). We found that the strength distribution of those connections underlying spike transfers matches the shape of the full connectivity, with a peak of very weak connections followed by a long tail (as in *Figure 1D*). Interestingly, the distribution of excitatory-to-excitatory spike transfers displays a secondary peak indicating an over-representation of strong connections (*Figure 3E*, top inset). By contrast, the absence of this secondary peak and the much higher primary peak (~1.5 M) for excitatory-to-inhibitory spike transfers suggest that inhibitory spikes are primarily the result of weak convergence (*Figure 3E*, bottom). The difference between the two populations is again greater at higher levels of activity.

This motif analysis allowed us to make several key predictions for how excitatory and inhibitory neurons participate in different types of motifs and with different synaptic strengths to propagate sequences. First, inhibitory neurons are usually activated by the convergent activation of multiple weak connections. The low fraction of inhibitory neurons and the high E-to-I connection probability (*Figure 1A*) makes inhibitory neurons strongly susceptible to spontaneous network activity. Consequently, we suggest that inhibitory neurons are less selective to their excitatory drivers, limiting their capacity to become followers. Second, we found that excitatory neurons are less susceptible to

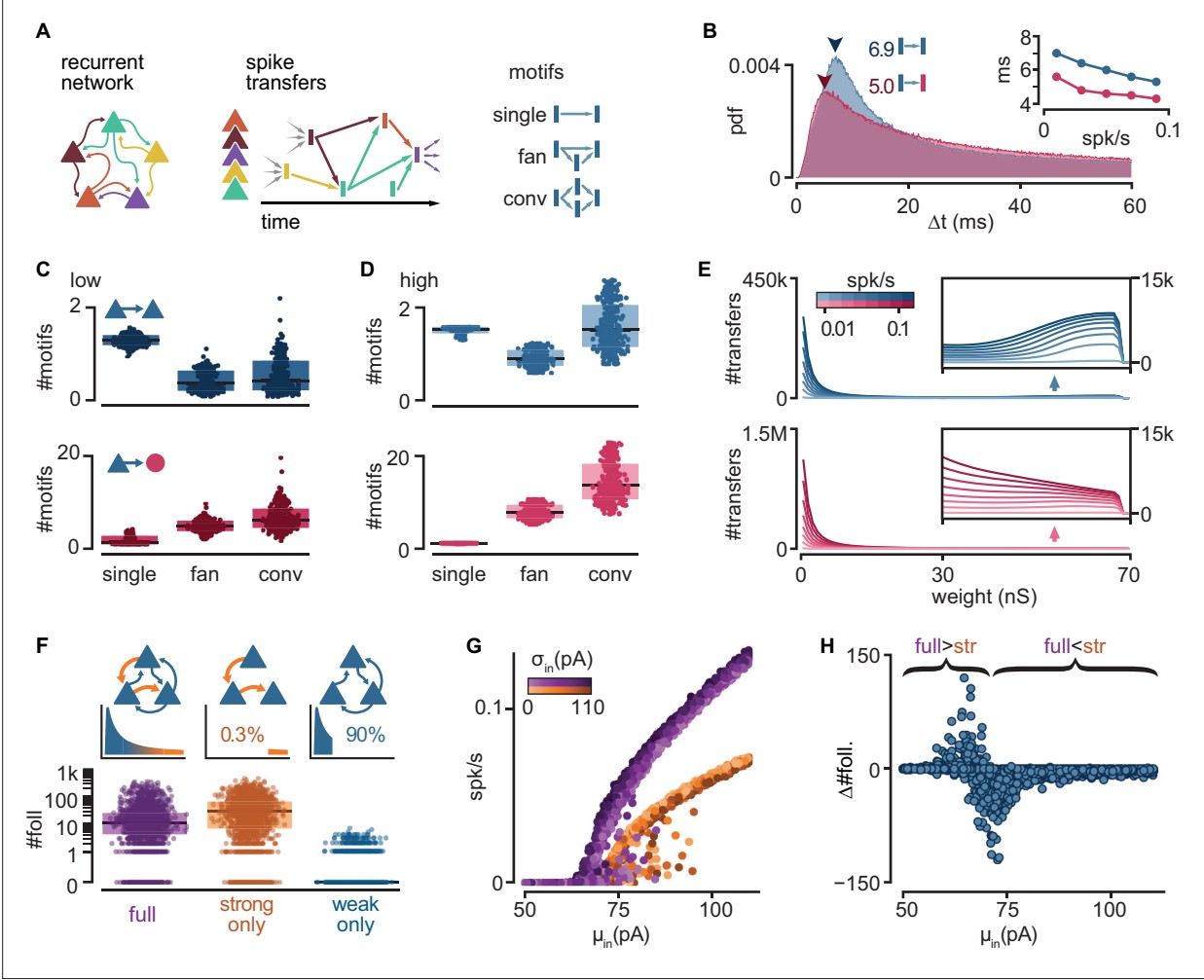

**Figure 3.** Connectivity underlying activity propagation in the network. (**A**) Schematic. Left: the recurrent network structure and spiking activity are combined to produce a directed graph of spike transfers. Right: motifs of spike transfers detected in this graph (conv: convergence). (**B**) Distribution of delays of excitatory-to-inhibitory (red) and excitatory-to-excitatory (blue) spike transfers. Average of all low-activity simulations ([0, 0.05] spk/s). Arrowheads: mode of each distribution. Inset: most common delay as a function of mean firing rate. (**C**) Number of motifs leading to an excitatory (blue) or inhibitory (red) neuron spike. Motifs defined in (**A**). Each dot represents the mean for all spikes in each simulation. Low-activity simulations. Boxes: median and [25th, 75th] percentiles. (**D**) Same as (**C**) for high-activity simulations. (**E**) Histograms of strengths of synapses leading to excitatory (blue) or inhibitory (red) neuron spikes. Insets: zoomed-in tails. Each line averages 25–100 simulations grouped in equal bins of mean firing rate (color bar). (**F**) Top: schematic of alternative network models and their truncated distribution of synaptic strengths. Bottom: number of detected excitatory followers per simulation (*n* = 2000 each). Boxes: median and [25th, 75th] percentiles. (**G**) Input–output curves for the full (purple) and strong-only (orange) models. (**H**) Difference between the number of followers detected in full and strong-only models under the same input. Brackets indicate regimes where the presence of weak connections increases (full >str) or decreases (full <str) the follower count. 2000 simulations.

The online version of this article includes the following figure supplement(s) for figure 3:

**Figure supplement 1.** Properties of full, strong-only, and weak-only models.

ongoing network activity than their inhibitory counterparts. Hence, single strong excitatory inputs are the dominant drive behind excitatory spikes and, thus, the likely conduit of sequence propagation.

To confirm the role of strong connections in sequence propagation, we built variations of our model in which we removed excitatory-to-excitatory connections according to their strengths (*Figure 3F*, top). We did not alter connectivity involving inhibitory neurons. We defined weak and strong connections as, respectively, the bottom 90% or top 0.3% of all connections, representing the two modes of the distribution of synaptic strengths of excitatory-to-excitatory spike transfers (*Figure 3E*, top). We found that networks with only strong connections result in very sparse excitatory connectivity, where 61.4% of excitatory neurons receive ≤2 excitatory connections (*Figure 1E*). We then re-ran a

random subset of all simulations using the modified models (*n* = 2000). We observed that networks with only weak connections require a much stronger drive than their intact counterparts to produce any activity (*Figure 3—figure supplement 1A*) and produce fewer excitatory followers (0% prob. of ≥10 followers, *Figure 3—figure supplement 1C*), in contrast with those with only strong connections (75%) (*Figure 3F*, bottom). Hence, we conclude that sparse connections strong enough to trigger a postsynaptic spike are necessary and sufficient to produce repeatable sequences in our model.

Interestingly, the model containing only strong connections often produces an excess of followers compared to the full model, suggesting that weak connections limit the reliability of those neurons under normal circumstances. Indeed, eliminating weak connections reduces the slope of the network's input/output (I/O) curve (*Figure 3G*). Consequently, networks with only strong connections remain at low firing rates for a broader range of inputs, in which more neurons behave as reliable followers (*Figure 2G*). Eliminating weak connections also causes a shift of the I/O curve to the right, increasing the range of inputs over which output activity is close to zero. This shift defines a narrow range of inputs where weak connections increase rather than decrease the number of followers (*Figure 3H*).

In summary, our model suggests that rare but strong and common but weak connections play different roles in the propagation of activity: the former promote reliable responses to single spikes, while the latter amplify spontaneous network activity and drive recurrent inhibition, effectively modulating the reliability of those responses.

## Sequences are composed of sub-sequences that correspond to sub-networks of strong connections

To better understand the regulation of sequential neuronal activations, we examined when and how sequences fail to propagate. We found that sequences could fail partially, with different sections of the same sequence failing to propagate independently of one another.

We chose a simulation with an intermediate firing rate (0.034 spk/s), and follower count (25) closest to the average for that firing rate. We observed that followers (*Figure 4A*, rows) can fail to activate over repeated trials with the same trigger (*Figure 4A*, columns). As a result, the exact spiking composition of a sequence can vary from one trial to the next, consistent with our definition of followers as neurons activated in a significant fraction of, but not necessarily all, trials. Because follower activations depend mainly on rare strong connections (*Figure 3F*), we reasoned that the activation or failure of any follower could affect the unfolding of a sequence. Consequently, we performed *k*-modes clustering of followers based on their activation over trials (*Figure 4B*, left). Our analysis reveals clusters composed of followers that tend to activate together in the same trials (*Figure 4C*, left). Here, we detected followers and performed clustering exclusively by activity, not connectivity. Nonetheless, when we mapped these follower clusters onto the graph of connections, we found that the connections within each cluster are strong and belong to the tail of the distribution of connection strengths (*Figure 4B, right, C, right*). We thus call each one of these follower clusters 'sub-networks' and their corresponding sections of the spiking sequence 'sub-sequences'. We observed similar decompositions into sub-sequences across simulations with different trigger neurons and numbers of followers (*Figure 4D*). These sub-networks persist despite random connectivity without hand-tuning of network structure. Across all simulations, strong connections (in the top 0.3% of the distribution) are more frequent *within* than *between* sub-networks, providing a mechanistic substrate for sub-network segregation (*Figure 4E*). Importantly, the few equally strong connections between sub-networks do not always guarantee the reliable propagation of activity between them due to unpredictable network effects resulting from recurrent interactions (*Figure 4F*).

To further examine the potential interdependence of sub-network activation, we selected the two largest sub-networks of every simulation (a and b) and classified each trial as one of four possible outcomes: full failure, a and b together, a alone, or b alone (*Figure 4G*, top). We defined a sub-network as activated if at least 40% of its followers fire at least once in that trial. We then extracted the entropy of these outcomes as a function of network mean firing rate across all simulations with at least two sub-networks (*Figure 4G*, bottom). The entropy is about 1 bit at low firing rates, indicating two equally likely outcomes (full failure or partial propagation). As spontaneous activity increases, entropy rises towards its maximum (2 bits), where all four outcomes become equally likely. This trend towards maximal entropy is consistent with recurrent connectivity and spontaneous activity in the model network being random, which, on average, avoids biases of activation of one sub-network over

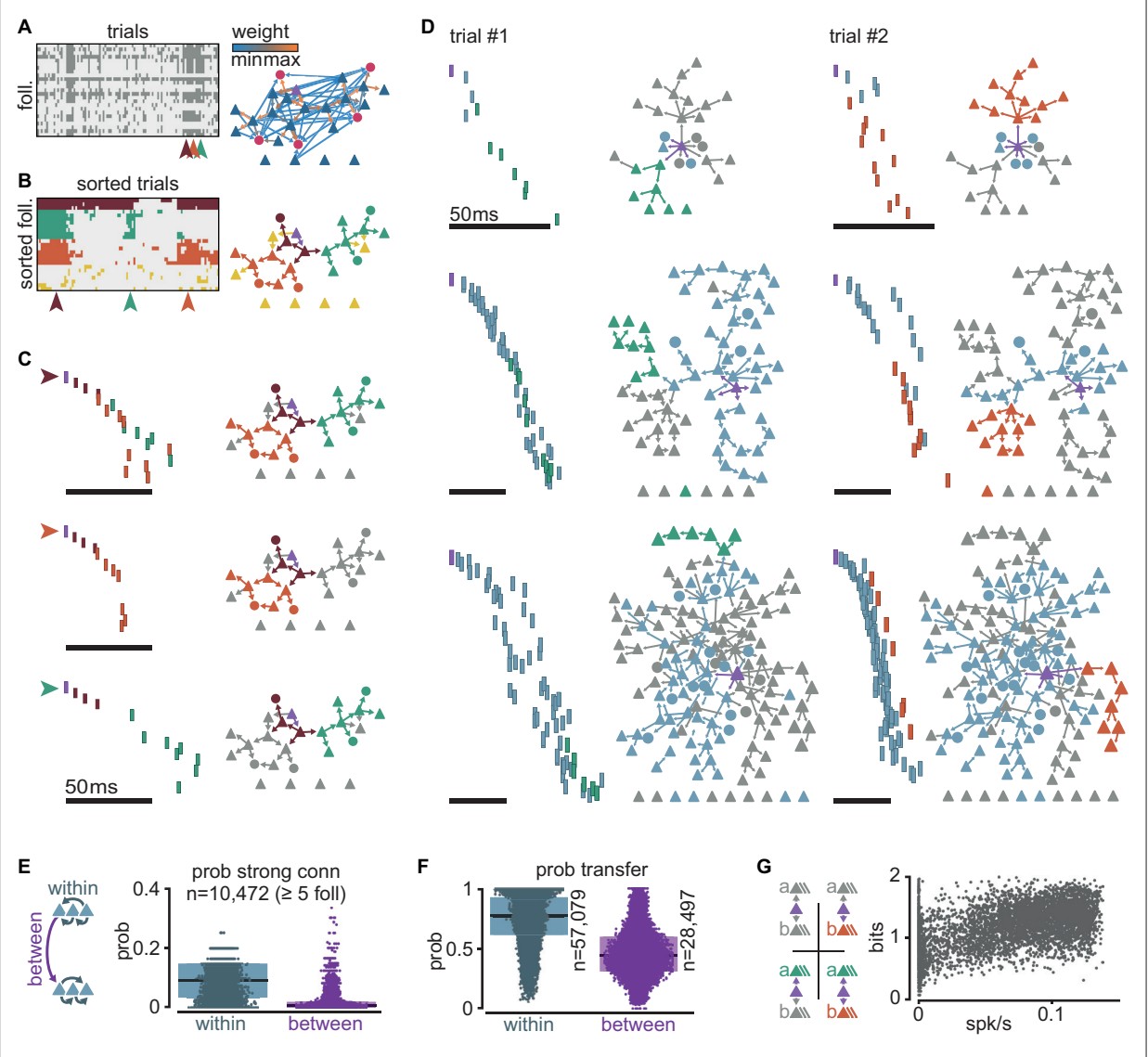

**Figure 4.** Clusters of sequential spikes reflect sub-networks of strongly connected followers. (**A**) Left: matrix of followers (rows) and trials (columns) of a representative simulation indicating follower activation (dark entries). Colored arrowheads indicate trials in (**C**). Right: graph of excitatory follower-to-follower connections colored by strength. Trigger neuron in purple. Neurons identified as followers but not directly connected are aligned below. (**B**) Left: matrix in (**A**) after sorting followers and trials according to k-modes clustering. Right: same graph as in (**A**) with only top 5% connections. Colors according to follower clustering. Note that members of orange, green, and brown clusters are connected to one another. (**C**) The three trials indicated in (**A**) and (**B**). Left: firing sequence with spikes colored according to follower cluster. Right: same graph as in (**B**) with inactive followers in gray. (**D**) Two trials (columns) of three different simulations to illustrate selective sub-network activation. Top to bottom: different trigger neurons with 28, 68, and 145 followers. Only the top 5% connections are shown. Followers were clustered as in (**B**). Green and orange: example active sub-networks. Blue: other active sub-networks. Gray: inactive followers. (**E**) Relationship between connectivity and follower clusters. Left: schematic illustrating connections between (purple) and within (blue) clusters. Right: estimated probability (measured frequency) of strong (top 0.3%) excitatory-to-excitatory connection within or between clusters for 10,472 clusters of at least 5 followers pooled across all 6000 simulations. (**F**) Estimated probability (measured frequency) of postsynaptic activation conditioned on presynaptic activation in the same trial, for excitatory-to-excitatory connections, pooled across all 6000 simulations. Boxes: median and [25th, 75th] percentiles. (**G**) Relationship between entropy and baseline firing rate. Left: schematic of four possible propagation scenarios in a simplified scheme where only two sub-networks are considered per simulation. Right: entropy over trials for each simulation with at least two sub-networks (n = 5371) as a function of network mean firing rate.

the other. In short, despite dense recurrent connectivity, the state of the network can turn on or off some sections of a sequence without altering other sections.

In summary, we found that sequence decomposition based on coactivity reveals sub-networks of strongly connected followers. The few strong connections linking different sub-networks do not always

reliably propagate activity. We predict that this apparent failure, in fact, adds flexibility, enabling the sub-networks to act as independent paths of propagation.

## Sparse external input can halt or facilitate the propagation of activity

We hypothesized that the connections between sub-networks could be critically important for sequence propagation. Indeed, the excitability and state of an entry point to a sub-network should affect sub-network activation, hence, we refer to these entry point neurons as 'gates'. By injecting a single extra spike to these gates, we could control the activation of subsequences and thus the path of propagation of the sequence.

We first identified the gates for every sub-network of every simulation based on activity, as the neurons within each sub-network with the shortest median spike delay from trigger.

To evaluate how the excitability of gate neurons might affect propagation in the rest of the sub-network, we varied the amount of excitatory or inhibitory conductances they received. We again ran a random selection of our original simulations ($n$ = 2000), but this time applying an additional spike from an external source to a randomly selected gate (*Figure 5A*). For the conductance of this external input, we chose the maximum value from either the excitatory or inhibitory synaptic conductance distribution (*Figure 1D*) while keeping the level of spontaneous activity in the rest of the network unchanged. We defined the level of activation of each sub-network ($a$) as the fraction of trials (out of 100) in which the sub-network is active and then computed the change of activation relative to control ($a_0$) under the altered conductance state of the gate ($\Delta a/a_0$, %) (*Figure 5B*). We found that external inputs to gate neurons are highly effective in controlling propagation through a sub-network: a single external inhibitory-input spike halves the probability of sub-network activation in 74% of the simulations and entirely halts its activation in 26% of the simulations. By contrast, a single external excitatory-input spike doubles baseline sub-network activation probability in 55% of the simulations (*Figure 5B*, *Figure 5—figure supplement 1A*).

To examine the effect of timing and strength of the external input on sub-network activation, we focused on the network in *Figure 4A*. We varied the sign (depolarizing vs. hyperpolarizing) and amplitude of the synaptic conductance of the external input to a gate neuron and its timing relative to trigger activation. We explored only conductances corresponding to single connections. We observed three different effects on the activation of the respective sub-network ($\Delta a/a_0$, *Figure 5C, D*). First, an inhibitory input of any strength in a temporal window of ~100 ms can acutely reduce sub-network activation, often halting propagation. Second, an excitatory input of intermediate strength can facilitate propagation. The temporal window of facilitation is centered on the expected spike time of the gate neuron. Lastly, an excitatory input of large amplitude can impair propagation if it occurs much earlier than the activation of the trigger neuron. This sub-network behavior results from the single-cell adaptive properties indicated by experimental data (*Hemberger et al., 2019*): the external input, by triggering the gate and some of the neurons in its sub-network, activates adaptive currents and thus raises their threshold for subsequent activation by the trigger neuron. We found equivalent effects on simulated networks with number of followers and baseline firing rates spanning orders of magnitude (*Figure 5—figure supplement 1B, C*). Computing the average across all tested sub-networks, we conclude that halting is most effective in a narrower time window (relative to trigger activation) than facilitation (70 and 100 ms, respectively) (*Figure 5E*, left). When considered relative to the expected spike time of the gate, we observed that the temporal windows are wider (*Figure 5F*, right). Importantly, while the halting window appears to have a sharp temporal boundary determined by the expected spike time of the gate in the sequence, the facilitation window increase with increasing input amplitude.

Finally, we examined how modulating the gate to one sub-network affects the activation of another (*Figure 5G, H*). Interestingly, this relationship between sub-networks is not symmetrical: in the single-network example, while orange sub-network activation is primarily independent of the state of the green sub-network (*Figure 5G*), gating of the orange sub-network unexpectedly facilitates green sub-network activation (*Figure 5H*), suggesting that the former typically exerts a net inhibitory influence on the latter. Indeed, strong excitatory inputs that might activate the orange sub-network early also reduce green sub-network activation (*Figure 5H*). More generally, we observed various complex interactions between pairs of sub-networks when examining other networks (*Figure 5—figure supplement 1B*, right). The sub-network interactions do not follow any particular trend (*Figure 5—figure*

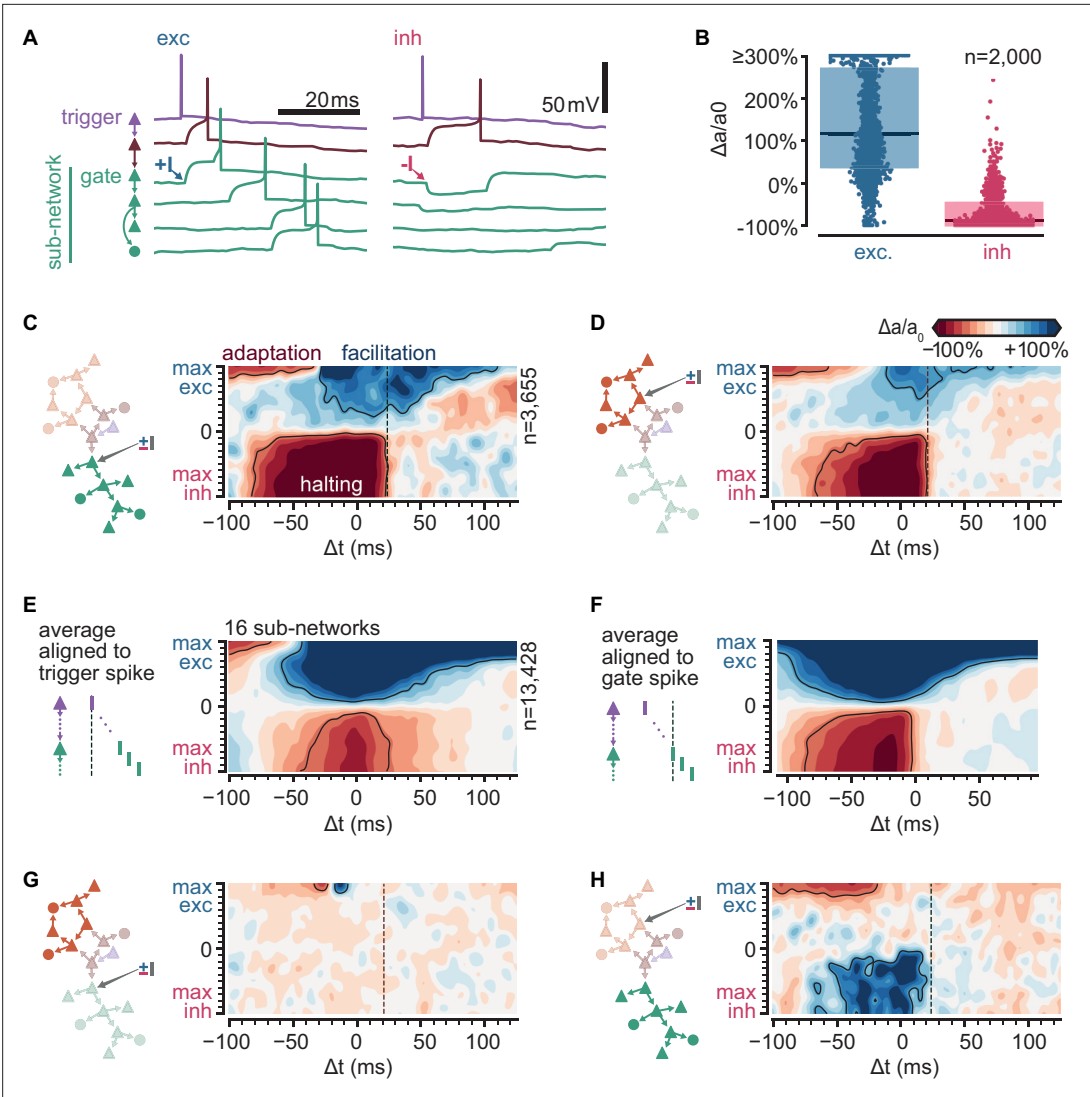

**Figure 5.** Sub-sequence activations depend on the state of gate neurons. (**A**) Voltage traces for six neurons in a sequence (network and colors as in *Figure 4B*) in two trials of simulations where the first neuron in the green sub-network receives a single excitatory (left,+I) or inhibitory (right, −I) input from an external source (arrows). (**B**) Fold change of sub-network activation above baseline ($\Delta a/a_0$, %) for sub-networks randomly selected across all 6000 original simulations. The model gate neuron of each sub-network received an additional single input from an external source at the beginning of each trial. (**C**) Schematic of protocol and map of the fold change in activation ($\Delta a/a_0$, %) for a sub-network when manipulating its gate (same network as *Figure 4A*). Arrow in schematic indicates stimulated gate. Solid outlines indicate combinations of strength and timing leading to halting (bottom, −50% $\Delta a/a_0$), facilitation (top right,+50% $\Delta a/a_0$), and adaptation (top left, −50% $\Delta a/a_0$). $\Delta t$ indicates the delay from trigger spike to external input. Dashed lines indicate the median spike time of the gate. (**D**) Same as (**C**) for a different sub-network. (**E**) Average map of the change in activation ($\Delta a/a_0$, %, same as **C**) computed from 16 sub-networks of 8 different networks spanning 5–462 followers and 0.01–0.1 spk/s baseline firing rate (networks in **C**) and (*Figure 5— figure supplement 1B*). Maps were aligned relative to trigger spike as in (**C**) before averaging. Solid outlines indicate ±50% $\Delta a/a_0$. Color bar as in (**C**). (**F**) Same as (**E**) but maps were aligned to each gate median spike time (dashed line in **C**) before averaging. (**G**) Same as (**C**) for the orange sub-network when manipulating the green gate. (**H**) Same as (**D**) for the green sub-network when manipulating the orange gate.

The online version of this article includes the following figure supplement(s) for figure 5:

**Figure supplement 1.** Gating sub-sequence activations.

*supplement 1C*), suggesting that they depend on the particular realization of the random connectivity in the network.

In sum, we identified gate neurons in different sub-networks whose activation is critical for the activation of the sub-network in a model based on the turtle cortex. External single spikes can control the state of gate neurons and thus halt or facilitate the activation of individual sub-networks. The effect of these spikes depends on their timing relative to the sequence trigger. Finally, the activation

of a sub-network may influence other sub-networks via recurrent connectivity leading to complex and non-reciprocal interactions.

## Sequences from multiple triggers reliably activate combinations of followers

To examine how sequences interact, we studied a new set of simulations where multiple-trigger neurons were activated simultaneously. We found that sequences from coactivated trigger neurons often interact, but they do so reliably, resulting in a combinatorial definition of followers.

We first simulated 100 trials by inducing a single spike from each of 2000 randomly selected trigger neurons in a representative network (as in *Figure 4A*, though see *Figure 6—figure supplement 1* for multiple network instantiations). Randomly pairing these trigger neurons, we estimated a very low probability that two excitatory neurons share followers (*Figure 6A, B*). Two triggers share at least 1 follower in only 3.12% of all tested pairs (*n* = 5000), and this probability drops to 0% for at least 11 followers.

Next, we generated a new simulation for each randomly chosen pair of trigger neurons while activating both simultaneously (100 trials). Comparing the followers resulting from these double-trigger activations to the followers from single-trigger activations reveals sets of followers (*Figure 6C*): ones responsive to the activation of only one of the triggers (a), followers responsive to the activation of only the other trigger (b), and followers responsive to their coactivation (a&b).

The number of followers obtained under a&b coactivation is smaller than the sum of the number of followers of a and b alone (*Figure 6D*, *Figure 6—figure supplement 1A*). Since a and b rarely share followers, the resulting sub-linear activation suggests frequent lateral inhibition, as we had observed between sub-networks (*Figure 5H*). Indeed, followers to a single trigger can be lost when that trigger is coactivated with another (a only and b only, *Figure 6E*, *Figure 6—figure supplement 1B*). Other followers, however, can emerge when two trigger neurons are coactivated but are absent in response to the activation of either trigger alone (ab only). We call these 'combination-specific followers' because their activation depends on whether multiple-trigger neurons are silent or active (a only, b only, and ab only). Combination-specific followers represent approximately half of all followers in the coactivation protocol (*Figure 6—figure supplement 1C*). Finally, some followers fire independently of whether the second trigger neuron is silent or active (a&ab and b&ab). We call these 'core followers'. We found that the excitatory and inhibitory compositions of core and combination-specific followers do not differ (*Figure 6—figure supplement 1D*). Core followers typically fire early in a sequence, whereas combination-specific followers become activated later (*Figure 6F*). This is consistent with experimental evidence that follower identity is more reliable early than late in a sequence (*Hemberger et al., 2019*), as captured by our model (*Figure 2D*). By contrast, the late activation of combination-specific followers suggests that modulation of sequences via intra-cortical interactions operates best on followers synaptically distant from the trigger neuron.

These findings prompted us to examine the effect of circuit interactions on sub-network activation in a known sequence. We used the trigger in the network in *Figure 4A* as our control with two known sub-networks and coactivated it with each of 2000 randomly selected excitatory neurons in the network, which we call 'contextual' neurons. By contrast with random spontaneous activity, these additional activations formed a context of activity that remained constant across trials. We then evaluated the effect of coactivation on the sub-networks of our control sequence (*Figure 6G, H*). We found that most contextual neurons excite or inhibit both sub-networks equally (diagonal trend, *Figure 6H*). Consistent with the tendency towards lateral inhibition (*Figure 6D*), 87.8% of the tested contextual neurons cause some inhibition of both sub-networks (bottom left quadrant, *Figure 6H*). Lateral excitation is also present but rarer (6.7%, top right quadrant, *Figure 6H*). The symmetry along the diagonal is likely a consequence of our random connectivity despite biological constraints and reflects our earlier observation that specific pairs of sub-networks can strongly influence each other, with the average effect being balanced (*Figure 5—figure supplement 1B*, right, *Figure 5—figure supplement 1C*). We saw that some contextual neurons prevent activation of one sub-network while facilitating the other (top left and bottom right quadrants, *Figure 6H*, 2.7%, and 2.9%, respectively, *Figure 6—figure supplement 1E*). The small percentage (~6%) of contextual neurons that can determine which sub-network becomes active suggests a highly specific form of routing. This result is consistent with our previous finding that excitatory followers generally tolerate ongoing network

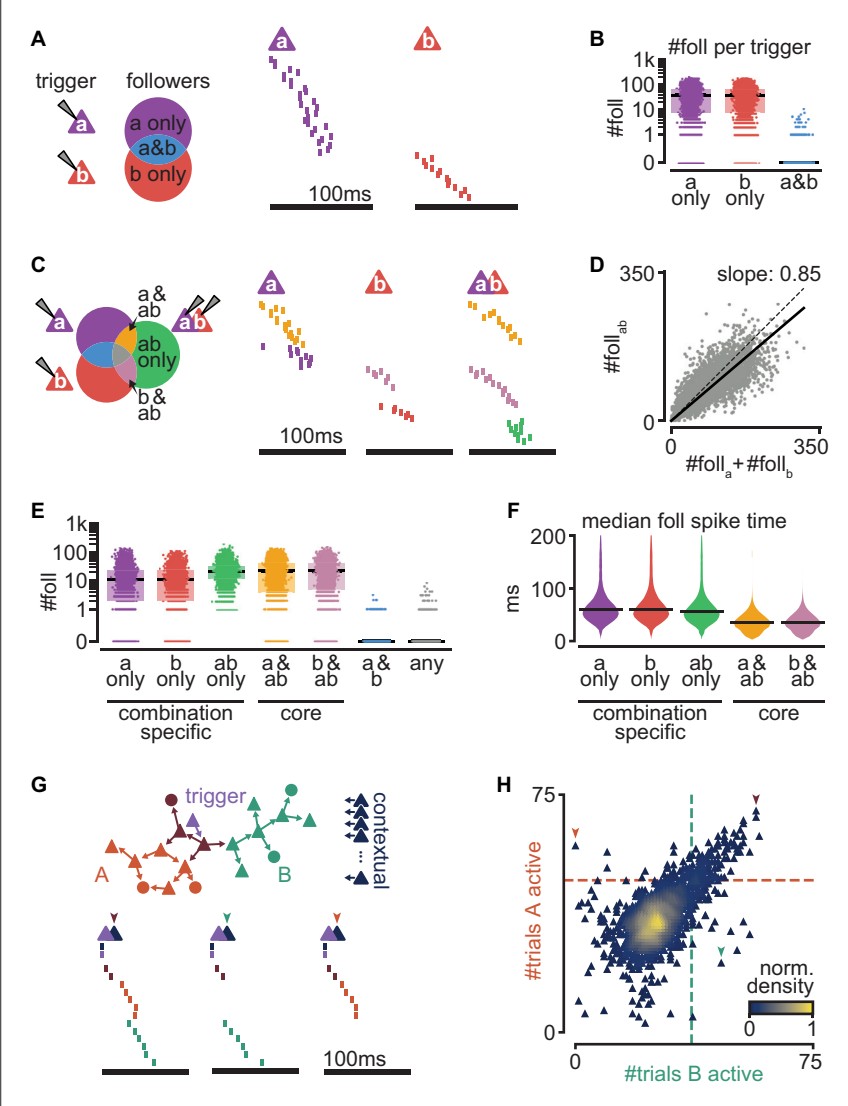

**Figure 6.** Interaction of sequences from multiple triggers reliably route activity. (**A**) Left: schematic of follower classes as a function of trigger neuron (purple: follower to a; red: follower to b; blue: follower to both a and b). Right: example sequences produced by the activation of a or b. Followers are sorted by trigger class (left) and median spike time. Triggers do not share any followers (blue trigger class). (**B**) Number of followers for each trigger class in (**A**) for all random pairs of simulations (*n* = 5000). Boxes: median and [25th, 75th] percentiles. (**C**) Left: schematic of follower classes as a function of trigger: single-trigger neuron (a or b) or simultaneous coactivation (ab). Blue, purple, and red as in (**A**). Right: same trials as in (**A**), and a trial under coactivation. Followers are sorted by trigger class (left) and median spike time. Same follower sorting in all trials. (**D**) Number of followers presents under trigger coactivation as a function of the sum of followers for single triggers. Dashed line: diagonal. Solid line: linear fit with zero intercept. (**E**) Number of followers for each trigger class in (**C**) for all random pairs of triggers (*n* = 5000). (**F**) Median follower spike time for each trigger class in (**C**) after pooling followers from all simulations (5000 simulations; 532,208 followers). Lines indicate median. (**G**) Top: trigger neuron (purple), followers, strong connections between them (arrows), and other neurons in the network (dark blue). Follower colors as in *Figure 4B*. Bottom: sequences triggered under simultaneous coactivation of trigger (purple) and one additional contextual neuron that facilitates propagation in both sub-networks (left), only in the green sub-network (middle), or only in the orange sub-network (right). (**H**) Effect on green and orange sub-network activation (G top) when the trigger neuron is coactivated with each of 2000 randomly selected contextual neurons. Dashed lines indicate baseline activation of each sub-network. Neurons are colored by local density. Colored arrows correspond to sequences in (**G**).

The online version of this article includes the following figure supplement(s) for figure 6:

**Figure supplement 1.** Interactions between sequences.

activity (*Figure 3C–E*). Given the size of our model network (2 × 2 mm, 100k neurons, *Figure 1A*), we estimate that in a given trial, the path of activation through the network can be influenced by the activation of ~5000 contextual neurons, enabling a big reservoir for computation through routing.

Together, our results indicate that overlap of sequences initiated by different triggers is rare but that interactions between the sequences are common. Importantly, we find that sequence interactions are reliable from trial to trial: they define groups of followers that activate reliably while being specific to the combined activation or silence of multiple triggers. While inhibitory interactions are most common, the activation of a sub-sequence can be facilitated through a small percentage of neurons within the recurrent network. Consequently, the state of just a few contextual neurons can selectively route cortical activity along alternative paths of propagation. Together, our results suggest that cortical computations may rely on reliable yet highly specific recurrent interactions between sequences.

## Discussion

We developed a model constrained by experimental data from the turtle visual cortex to investigate the effect of single spikes in recurrent networks with otherwise random connectivity. Our model produces reliable sequences from single spikes as experimentally measured ex vivo and predicts their existence in vivo in the turtle cortex, where firing rates are higher, with marked differences between excitatory and inhibitory neurons. We found that strong but rare connections form a substrate for reliable propagation in the network, while dense but weak connections can enhance or disrupt propagation. Our model reveals that sequences may be decomposed into multiple sub-sequences, each reflecting a sub-network of strongly connected followers, which emerge without any hand-tuning of network structures. We showed that activation of individual sub-sequences is sensitive to the state of a few gate neurons and can thus be controlled through ongoing recurrent activity or external inputs. We observed that different trigger neurons rarely share followers, but their sequences often interact, suggesting that downstream circuits may access the combined information from different sources. Indeed, the selectivity of followers to the combinatorial activation of multiple triggers suggests a mechanism that can produce a wide repertoire of activations while remaining reliable and specific. Finally, we found that the exact path of propagation is influenced by contextual activity provided by the activation of a small percentage of other neurons in the network. In summary, our biologically constrained model allows us to dissect the mechanisms behind the reliable and flexible propagation of single spikes, yielding insights and predictions into how cortical networks may transmit and combine streams of information.

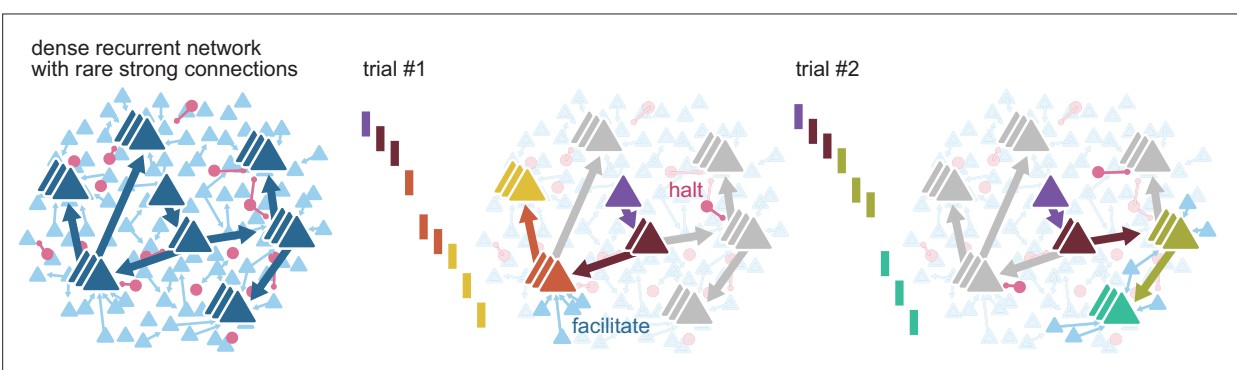

**Figure 7.** Routing using sparse strong connectivity. As activity propagates in cortex, a spike will preferentially propagate through strong connections. The strength of those connections ensures reliable propagation, while their sparsity enables flexibility, creating sub-networks with gates where propagation can be halted or facilitated. The halting of activity can appear as partial failures of the expected sequence of spikes over a different history of neuron activations. The spiking state of other neurons in the recurrent network, together with external inputs, form a context that determines the final path for each spike.

## Routing

The sequences in our model represent a form of propagating activity. They are related to, but distinct from, previous frameworks: temporary breaking of excitatory/inhibitory balance, synfire chains, neuronal avalanches, and stochastic resonance.

Our work suggests that the disruption of propagating spiking sequences may be a hallmark of flexibility, where apparent failures to propagate activity are the consequence of routing it (*Figure 7*). For instance, changes in contextual activity across trials may frequently route propagation away from certain neurons, yielding what we would identify as followers with low firing rate modulation. Different replay sequences of place cells in the same environment have been observed in rat hippocampus and linked to flexible planning in goal-directed and memory tasks, and our model provides a plausible circuit mechanism for this flexible trajectory sequence generation (*Pfeiffer and Foster, 2013*; *Widloski and Foster, 2022*). We propose that the dynamic routing of activity relies on two elements: the existence, for each propagation path, of few points of failure and specific mechanisms to manipulate them. The strength of some connections in our model enables reliable propagation, while their sparsity defines groups of neurons with few entry connections (gates). Precisely timed external inputs can selectively open or close these gates. In addition, ongoing parallel sequences can also influence the path of propagation in the form of lateral inhibition (competition) or excitation (cooperation) between sub-networks in the recurrent network. We expect that complex computations can be built from this proposed form of gating. Previous theoretical work on signal propagation in recurrent networks with irregular background activity found that the combination of parallel pathways and gating can implement different logic statements, illustrating how gating can form the basis for more complex computations (*Vogels and Abbott, 2005*). Indeed, recent advances in machine-learning result from leveraging gating to control the flow of information more explicitly. For instance, gating can implement robust working memory in recurrent networks (*Chung et al., 2014*), prevent catastrophic forgetting through context-dependent learning (*Masse et al., 2018*), and produce powerful attention mechanisms (*Vaswani et al., 2017*).

Although a single spike can reliably trigger many others in the network, we never observed explosive amplification of activity in our model. On the contrary, multiple excitatory spikes result in a sub-linear summation of responses (*Figure 6D*), likely reflecting the frequent lateral inhibition onto sub-sequences (*Figure 6H*). Together, this suggests that sequences operate in a regime where local inhibition often balances excitatory signals. We did not explicitly construct our model to display an excitatory/inhibitory balance. Still, this form of lateral inhibition and routing via the alteration of excitatory and inhibitory conductances in gate neurons are reminiscent of 'precise balance', a form of balance within short temporal windows (*Bhatia et al., 2019*). Hence, our work is consistent with and extends previous theoretical work proposing that the gating of firing rate signals could be arbitrated via the disruption of excitatory/inhibitory balance in local populations (*Vogels and Abbott, 2009*).

Synfire chains are a classical model of reliable propagation where pools of 100–250 neurons are connected sequentially in a divergent–convergent pattern. Near-synchronous activation of one pool can thus guarantee postsynaptic spikes in the next pool even under partial failure, so reliable synchronous activation recovers in later pools (*Abeles, 1991*; *Diesmann et al., 1999*; *Kumar et al., 2010*). While the biological evidence for synfire chains is inconclusive (*Abeles et al., 1993*; *Fiete et al., 2010*; *Long et al., 2010*; *Oram et al., 1999*; *Prut et al., 1998*), theoretical studies have embedded them in recurrent networks yielding insights into the specific network structures needed for effective propagation and gating of activity (*Aviel et al., 2003*; *Kremkow et al., 2010*; *Kumar et al., 2010*; *Mehring et al., 2003*). A related structure to synfire chains is that of polychronous chains, where synchrony is replaced by precise time-locked patterns that account for the distribution of synaptic delays (*Egger et al., 2020*; *Izhikevich, 2006*). These time-locked patterns, or polychronous groups, self-reinforce through spike-timing-dependent plasticity, resulting in stronger synapses. Contrary to classical synfire chains, connectivity in our network is unstructured, and activity propagates over synaptic chains that are sparse (≤2 connections, *Figures 2E and 3F*) yet powerful enough to make the synchronous firing of large groups of neurons unnecessary. This sparsity enables the selective halting of propagation by even small inputs (*Figure 5A*) or sequence-to-sequence interactions (*Figure 6C*) and causes late followers to be the most prone to modulation, that is, least reliable (*Figures 2D and 6H*). Interestingly, the unstructured random connectivity in our model captures very well the experimental finding that 80% of excitatory neurons trigger sequences even when activated in isolation (*Hemberger et al.,*

*2019*; *Figure 2G*), suggesting that sparse chains of strong connections are indeed common, at least in turtle cortex. Consequently, reliable propagation may involve far fewer neurons and spikes than initially predicted by synfire models, with significant experimental and computational implications.

In our model, a single spike could trigger multiple ones without runaway excitation, suggesting that sequences share some properties with neuronal avalanches (*Beggs and Plenz, 2003*). Criticality, a dynamical regime that can maximize information transmission in recurrent networks (*Beggs, 2008*), has been associated with the ex vivo turtle cortex, from which data constrained our model, after observing scale-free statistics of population activity (*Shew et al., 2015*). Although cortical criticality does not require reliability, it does not preclude it. Indeed, repeatable and precise LFP avalanches have been found in slice cultures, possibly reflecting the existence of underlying repeatable spiking sequences (*Beggs and Plenz, 2004*). Our finding of spike sequences as the expression of routing by strong connections may thus be an alternative view, compatible with the hypothesis of cortical criticality. Importantly, we interpret the halting of sequences as a form of contextual gating. While avalanches may statistically describe sequence sizes as a stochastic branching process, our simulations with fixed external input (*Figure 5*) and contextual activity (*Figure 6*) suggest that reliable control of sub-sequences is possible. Consequently, network states that remain constant from trial to trial will produce sequence statistics that differ from pure branching-process statistics (e.g., powerlaws). Experimentally, this might be achieved pharmacologically or by increasing the number of trigger neurons that are coactivated (*Figure 6G, H*).

Lastly, stochastic resonance is a phenomenon where noise can enhance the response of a system (*Collins et al., 1995a*). In particular, long-tailed distributions of strengths may create the opportunity for aperiodic stochastic resonance between weak and strong spike inputs (*Collins et al., 1995b*; *Teramae et al., 2012*). Indeed, our simulations with and without weak connections reveal narrow ranges of input where weak connections could increase or decrease reliability (*Figure 3H*). This global control mechanism contrasts with the selectivity of single sub-networks to the activation of a few neurons within the recurrent circuit (*Figure 6H*). As with criticality and synfire chains, our results suggest that some routing mechanisms might not be stochastic and operate at the population level, but rather that they are reliable and fine-grained.

## Connectivity

Our modeling results show that rare but strong connections are key to the generation of sequential activity from single spikes. Such sparse and strong connections are a common feature of brain connectivity. Indeed, long-tailed distributions of connection strengths are found in rat visual cortex (*Song et al., 2005*), mouse barrel and visual cortices (*Cossell et al., 2015*; *Lefort et al., 2009*), rat and guinea pig hippocampus (*Ikegaya et al., 2013*; *Sayer et al., 1990*), and human cortex (*Shapson-Coe et al., 2021*). Modeling and in vitro studies in hippocampus and cortex have suggested that sparse strong inputs substantially affect their postsynaptic partners and network dynamics (*Franks and Isaacson, 2006*; *Ikegaya et al., 2013*; *Setareh et al., 2017*). Furthermore, repeatable sequences of action potentials have also been reported in most of these mammalian cortical structures (*Buzsáki and Tingley, 2018*; *Carrillo-Reid et al., 2015*; *Dechery and MacLean, 2017*; *Diba and Buzsáki, 2007*; *Dragoi and Tonegawa, 2011*; *Luczak et al., 2015*; *Luczak and Maclean, 2012*; *Vaz et al., 2020*). Thus, could strong connections underlie cortical sequences more generally, and are sequences an ancient feature of cortical computation? We based our model on data from the turtle cortex, yet other species may introduce variations that might be relevant, including differences in operating firing rates or neuronal populations. For instance, the reliability of sequence propagation may depend on connectivity features we did not explore here, such as the presence of different connectivity motifs or differences in the distance-, layer-, and population-specific connection probabilities (*Jiang et al., 2015*; *Song et al., 2005*). Further modeling and experimental work on the cortical responses of diverse species will be needed to provide a comprehensive comparative perspective (*Laurent, 2020*).

Our study identified distinct roles for strong and weak connections, providing reliability and flexibility, respectively. Experimental distributions of trial-to-trial variability of EPSP sizes in rat and mouse cortices show that the amplitude of large EPSPs is less variable than that of small EPSPs, supporting the greater reliability of strong connections (*Buzsáki and Mizuseki, 2014*; *Ikegaya et al., 2013*; *Lefort et al., 2009*). Recent electron microscopy evidence from mouse primary visual cortex (V1) shows that the distribution of synapse sizes between L2/3 pyramidal neurons is best described by the

combination of two lognormal distributions (*Dorkenwald et al., 2019*). This binarization of excitatory synapse types may be the morphological equivalent of our modeling predictions.

Although we based many model parameters on experimental data from the turtle visual cortex (connection strengths, single-neuron properties, connection probabilities), our model has otherwise random connectivity. Consequently, our model displays a high degree of heterogeneity, with sub-networks emerging without fine-tuned connectivity, allowing us to make general predictions about sequence propagation and the impact of single spikes. However, other features of cortical organization, such as structured connectivity, neuronal diversity, and dendritic interactions, might affect activity propagation and routing. Indeed, certain nonrandom features may be expected in turtle cortical connectivity, such as axonal projection biases (*Shein-Idelson et al., 2017*), differences between apical and basal dendritic connectivity, gradients of connectivity across the cortical surface (*Fournier et al., 2015*), or plasticity-derived structures. Transcriptomic and morphological evidence suggests that turtle cortex contains diverse neuronal types, as seen in mammalian cortex (*Nenadic et al., 2003*; *Tosches et al., 2018*). Different neuronal types may play different roles in information routing. For instance, somatostatin- and parvalbumin-expressing neurons in mouse V1 may play different roles in controlling trial-to-trial reliability (*Rikhye et al., 2021*). Finally, our model represents neurons as single compartments, but complex input interactions and nonlinearities may occur at the level of dendritic arbors. For instance, synapses may cluster in short dendritic segments to effectively drive postsynaptic spiking (*Kirchner and Gjorgjieva, 2021*; *Scholl et al., 2021*); these inputs may be locally balanced by inhibition *Iascone et al., 2020*; and sequential activation within clusters may be particularly effective at triggering somatic spiking (*Ishikawa and Ikegaya, 2020*). The consequences of these aspects of cortical organization on information routing via strong connections remain to be explored.

## Function

Evidence from rodents has often associated spiking sequences with memory or spatial tasks (*Buzsáki and Tingley, 2018*; *Diba and Buzsáki, 2007*; *Dragoi and Tonegawa, 2011*; *Modi et al., 2014*; *Vaz et al., 2020*) or reported sequences within the spontaneous or stimulus-evoked activity in sensory areas (*Carrillo-Reid et al., 2015*; *Dechery and MacLean, 2017*; *Fellous et al., 2004*; *Luczak et al., 2015*; *Luczak and Maclean, 2012*). More generally, spiking sequences have been proposed as parts of an information-coding scheme for communication within cortex (*Luczak et al., 2015*) or as content-free structures that can reference experiences distributed across multiple cortical modules (*Buzsáki and Tingley, 2018*). Although spking sequences were observed in the ex vivo turtle cortex, we predict the existence of sequences under in vivo levels of baseline activity. We propose that reliable routing of sequential activity might implement the visual tuning of cortical neurons. The dorsal cortex of turtles is a visual processing area that shows no clear retinotopy (*Fournier et al., 2018*) but displays stimulus-evoked wave patterns (*Nenadic et al., 2003*; *Prechtl et al., 1997*). Most neurons have receptive fields covering the entire visual field and display a wide range of orientation selectivity (*Fournier et al., 2018*). The presence of orientation tuning in the turtle cortex and our result that excitatory neurons are mainly activated via strong connections (*Figure 3C–E*) predict that like-tuned neurons should share strong connections. Interestingly, long-tailed distributions of connection strengths between L2/3 neurons of mouse V1 play a prominent role in orientation tuning in vivo (*Cossell et al., 2015*) (but see opposite evidence in ferret V1 *Scholl et al., 2021*). In mouse V1, stimulus presentation triggers broadly tuned depolarization via a majority of weak connections, while sparse strong connections determine orientation-selective responses (*Cossell et al., 2015*). This untuned depolarization via weak connections is consistent with our finding of activity regimes where weak connections can modulate the reliability of propagation (*Figure 3H*). Overall, the dual roles in routing played by weak and strong connectivity may have corresponding roles in what we describe as tuning. 'Tuning' is an operational term describing a slice of the recurrent pipeline of transformations linking stimulus to perception or behavior. Routing of spikes and tuning thus likely represent different ways of describing the same processes.

Our model goes beyond generating reliable sequences from single spikes in many other ways. For example, experiments in the turtle cortex with multiple-trigger neuron activation revealed non-additive effects on follower composition (*Hemberger et al., 2019*). We can now mechanistically explain these as interactions between sub-networks of strong connections. In our model, the scale of the temporal windows for effective single-sub-network gating is about twice as long as the standard

deviation of follower spike times (approximately 100 and 50 ms, respectively, *Figures 2H and 5C*). Consequently, parallel running sequences (e.g., as observed in the mouse auditory cortex *Dechery and MacLean, 2017*) may have the temporal precision to interact consistently across trials. Indeed, our model produces late-activated followers that are reliable and specific to the coactivation of two trigger neurons (*Figure 6E*). The presence of combination-specific followers in our model suggests that the tuning of a neuron may be expressed as the result of logical operations on the tuning properties of other neurons in the recurrent circuit. Under this view, lateral inhibition between sub-networks, mediated through the convergence of weak connections (*Figure 3D, E*), may perform nontrivial computations on the properties of stimuli in a sensory cortex. Indeed, lateral inhibition triggered by early responding principal neurons implements concentration-invariant coding in mouse olfactory cortex (*Bolding and Franks, 2018*). In line with our observation of common lateral inhibition between sub-networks (*Figures 5H and 6H*), activations of single neurons in mouse V1 tend to inhibit neurons with overlapping tuning (feature competition) (*Chettih and Harvey, 2019*). Similarly, highly specific lateral excitation between sub-networks (*Figure 6H*) may act as a mechanism to bind together stimulus properties (*Abeles et al., 2004*). Given the density of excitatory neurons and the frequent presence of strong connections (*Figure 1A, E*), such circuits may display a high combinatorial capacity (*Bienenstock, 1995*).

In conclusion, our results establish a mechanistic link between three elements common to most cortical circuits: sparse firing, spiking sequences, and long-tailed distributions of connection strengths. We provide a computational interpretation for spiking sequences as the expression of reliable but flexible routing of single spikes over sets of neurons connected by strong connections (*Figure 7*). By dissecting how a cortex responds to single spikes, we illustrate the relevance of this elementary unit of communication between neurons in cortical computations.

**Table 1.** Neuron and synapse model parameters. Asterisk (*) indicates parameters fitted from experimental data (*Hemberger et al., 2019*). Parameters of excitatory synaptic conductance are for a truncated lognormal distribution.

| Variable | Value |
| --- | --- |
| Excitatory reversal potential ($E_e$) | 10 mV |
| Inhibitory reversal potential ($E_i$) | −75 mV |
| Synaptic conductance time constant* | 1.103681 ms |
| Excitatory synaptic conductance (mean)* | 3.73 nS |
| Excitatory synaptic conductance (std)* | 6.51 nS |
| Excitatory synaptic conductance (max)* | 67.8 nS |
| Leak reversal potential ($E_L$) | −70.6 mV |
| Spike detection threshold | 0 mV |
| Membrane reset potential | −60 mV |
| Spike initiation threshold ($V_T$) | −50.4 mV |
| Membrane capacitance* ($C$) | 239.8 pF |
| Leak conductance* ($g_L$) | 4.2 nS |
| Subthreshold adaptation ($a$) | 4 nS |
| Spike-triggered adaptation ($b$) | 80.5 pA |
| Adaptation time constant ($\tau_w$) | 144 ms |
| Slope factor ($\Delta_T$) | 2 mV |
| Refractory period | 2 ms |

## Methods

### Neuron and synapse models

We simulated neurons as adaptive exponential integrate-and-fire (see *Table 1* for parameters) (*Brette and Gerstner, 2005*). We used the NEST model 'aeif_cond_exp' (see Methods, Simulations), which implements a conductance-based exponential integrate-and-fire neuron model where the membrane potential is given by:

$$C\frac{dV}{dt} = -g_L(V - E_L) + g_L\Delta_T exp\frac{V - V_T}{\Delta_T} - g_e(V - E_e) - g_i(V - E_i) - w + I_e$$
$$\tau_w\frac{dw}{dt} = a(V - E_L) - w$$

At each firing time the variable $w$ is increased by an amount, which accounts for spike-triggered adaptation. Inhibitory and excitatory synaptic conductances ($g_i$ and $g_e$) are implemented as exponential decay kernels (see NEST documentation, *Linssen et al., 2018*).

To estimate membrane properties, we used least squares to match the output of our neuron model to the membrane potential of real neurons in whole-cell clamp under different levels of current injection (*Hemberger et al., 2019*). An example of the fit can be seen in *Figure 1B*. We first fit the model membrane leak conductance (resistance) using the steady state of the experimentally measured membrane potential after current injection. We then fit the model membrane capacitance (which determines the membrane time constant) using the experimentally measured potential immediately after the onset of current injection. We fitted all traces independently ($n = 3886$), and used the median of the distributions for all neurons in the model. Membrane parameters resulted in a rheobase current of 150 pA for a model neuron in isolation.

We fitted synaptic time constants to rise times of EPSPs ($n = 382$) obtained in paired-patch recordings (same as *Figure 1D*; *Hemberger et al., 2019*) using least squares and used the median for our model synapses. We converted experimentally measured EPSP amplitudes (same $n = 382$) to synaptic conductances (*Figure 1D*) using least squares and fitted a lognormal distribution via maximum likelihood estimation. When drawing strengths from the resulting lognormal distribution, we discarded and re-sampled conductances higher than the maximum conductance of all fitted EPSP amplitudes (67.8 nS, 21 mV). Inhibitory synaptic conductances were drawn from the same truncated lognormal distribution and scaled up by 8, resulting in currents 2.5 times stronger and a maximum IPSP amplitude of −21 mV on an isolated model neuron that has been depolarized to 50 mV.

## Network

Our network model consisted of 93,000 excitatory neurons and 7000 inhibitory neurons that were randomly assigned a location on a square of side 2000 μm.

Neurons were connected randomly. The probability of connection between two given neurons changed as a function of Euclidean distance in the plane, assuming periodic boundary conditions. We first estimated connection probability at regular distance intervals from 918 paired whole-cell patch-clamp recordings (*Figure 1C*; *Hemberger et al., 2019*). We then fitted these estimates with a Gaussian profile using least squares. Finally, we scaled the profile so that the expected probability within a 200-μm radius matched population-specific probabilities measured experimentally (*Figure 1A*). The degrees of connectivity were not fixed, so inhomogeneities in neuron location and randomness of connectivity meant that neurons received a variable number of connections. For instance, the distributions of numbers of E–E weak and strong connections are given in *Figure 1E*.

All strengths and delays were assigned independently for each connection. Neurons did not have autapses.

## Simulations

Simulations were run using NEST 2.16 (*Linssen et al., 2018*) with a time step of 0.1ms. Each simulation containing 100 trials took between 30 and 45 min on an Intel(R) Xeon(R) Gold 6152 CPU @ 2.10 GHz.

We instantiated 300 networks and generated 20 simulations of each instantiation, yielding a batch of 6000 simulations. Each neuron in each simulation received a random current sampled independently every 1 ms from a Gaussian distribution $N(\mu_{in}, \sigma_{in})$. Each simulation had randomly assigned input parameters ($\mu_{in}$ and $\sigma_{in}$). Additionally, a single-trigger neuron was randomly chosen in each network and driven to spike at regular intervals (see Methods, Definition of followers). Whenever we re-ran a simulation, we used the same input statistics and trigger neuron unless otherwise stated.

Certain mean input levels ($\mu_{in}$) could lead to self-sustaining activity (*Figure 1H*) but only after a few spikes had occurred in the network, possibly resulting in different activity early and late in a simulation. Consequently, we kick-started all our simulations with a volley of spikes sent to a random selection of 500 excitatory neurons in the first 100 ms and then discarded the initial 1000 ms.

We analyzed all 6000 original simulations to identify sequences of spikes (*Figure 2*) and sub-sequences (*Figure 4*). A random selection of 900 simulations was analyzed to extract connectivity motifs (*Figure 3A–D*). A random selection of 2000 simulations was re-run with truncated connectivity (*Figure 3F–H*). To study the effect of external spikes on sub-sequences (*Figure 5A, B*), we re-ran a random selection of 2000 simulations where the gate of one randomly chosen sub-network received an additional external spike. To study how external spike parameters affected sub-sequences (*Figure 5C–H*, *Figure 5—figure supplement 1*), we selected 8 simulations and re-ran each one between 1142 and 3655 times while varying external spike strength and timing. To study the effect of

coactivation of multiple-trigger neurons (*Figure 6*), we selected a representative simulation (same as *Figure 4A*) and re-ran it 7000 times with new randomly chosen triggers (but same input parameters, $\mu_{in}$, and $\sigma_{in}$). We repeated this analysis on another 50 randomly chosen simulations re-ran 800 times each (*Figure 6—figure supplement 1*).

## Simulation firing rates

We focused on capturing the mean firing rates from ex vivo and in vivo spontaneous activity ([0, 0.05] and [0.02, 0.09] spk/s, *Hemberger et al., 2019*) because they provide a clear baseline of uncorrelated variability for new incoming spikes. To compute the mean firing rate of each simulation, we took the average during the baseline periods (100 ms before each spike injection). Our averages include all neurons (excitatory and inhibitory) equivalent to a 2 × 2 mm slab of tissue (*Figure 1A*). Since experimental estimates are limited to those high-rate neurons that can be accessed and spike sorted in extracellular recordings (*Fournier et al., 2018*; *Hemberger et al., 2019*), comparing them to our model is not trivial. Along those lines, most excitatory neurons in our model produced none-to-few spikes (*Figure 2—figure supplement 1E*), and excluding them increases mean rates 10-fold (to 0.22–1.04 spk/s).

As well as low average firing rates, our model also produces fluctuations of higher instantaneous firing rates reminiscent of the waves observed experimentally (*Prechtl et al., 1997*; *Shew et al., 2015*; *Figure 2—figure supplement 1C, D*). Convolving spike trains in our simulations with a 250-ms Gaussian window yields peak instantaneous firing rates one order of magnitude higher than their average (maximum firing rate per neuron, averaged across all neurons, 0.53–1.36 exc. spk/s, 1.7–3.7 inh. spk/s), and comparable to those observed in vivo under visual stimulation (*Fournier et al., 2018*).

## Definition of followers

Every 400 ms, for 100 trials, we set the voltage of the trigger model neuron above the spike detection threshold, generating an instantaneous spike (trials). For each trial, we defined a window before spike injection (100 ms) and a window after spike injection (300 ms). We computed the mean firing rate for all windows before and after spike injection for all other neurons in the network and defined the difference in firing rate before and after as the firing rate modulation of each neuron (ΔFR). We normalized ΔFR by dividing it by that expected from a neuron that never spiked before spike injection and that spiked precisely once after spike injection.

We defined a null distribution for firing rate modulation (ΔFR) as the difference of two random samples from two Poisson random variables with the same underlying rate. The rates of the two Poisson random variables were adjusted to account for the different duration of the time windows before and after spike injection. The underlying rate was taken as the mean firing rate of all model neurons in the windows before spike injection. The resulting null distribution was centered at zero but displayed a non-zero variance. We used $p < 10^{-7}$ as an upper boundary to the null distribution to segregate spurious firing rate increases from reliable neurons. Since we tested all $10^5$ neurons in our model (except the trigger neuron), we expected one false positive on average every 100 simulations. Due to big differences in firing rates, we computed different null distributions for the excitatory and inhibitory populations. Due to the very low baseline firing rates in our networks, we could not detect the opposite of followers: neurons that decreased their firing rate during our spike induction protocol. However, our analysis of the coactivation of multiple triggers (*Figure 6*) showed that such neurons do exist.

A higher number of trials per simulation would have allowed a better estimation of ΔFR and the detection of extra followers. However, we found that 100 trials detected the most reliable followers while limiting simulation times (see Methods, Simulations) and keeping our protocol and predictions experimentally relevant.

## Lognormal fit bootstrap

We generated 50,000 bootstrapped samples of the probability that a neuron had at least one strong excitatory-to-excitatory connection (*Figure 1F*). A connection was strong if it fell between 50.6 and 67.8 nS, corresponding, respectively, to the top 99.7% of the original fitted lognormal distribution (*Figure 1D*) and the maximum possible strength.

The number of strong connections $s_{ij}$ of a model neuron $i$ in bootstrap step $j$ follows a Binomial distribution $B(n_i, p_j)$, where $n_i$ is the total number of connections of neuron $i$, and $p_j$ is the probability that a connection falls within the strong weight range. To estimate $p_j$ we sampled 122 times, with replacement, from a set of 122 experimentally obtained EPSP amplitudes and fitted a new truncated lognormal. Our model neurons received a variable number of connections $n_i$ (see Methods, Network), which was well described by a discretized Gaussian distribution $N(\mu = 745, \sigma = 27)$. For each bootstrapped step $j$, we thus generated 1000 values of $n_i$, drew one sample $s_{ij}$ of each $B(n_i, p_j)$, and calculated the ratio of all $s_{ij}$ bigger or equal to 1.

### Entropy of follower rank

To compare the reliability of the order of follower activation to that seen in experiments, we used the same entropy-based measure as used first in experiments (*Hemberger et al., 2019*; *Figure 2D*). For every trial, followers were ordered by the time of their first spike and we calculated the entropy of follower identity for each rank:

$$H_k = -\sum_i^n p_{ik}\log_2 p_{ik}$$

where $n$ is the number of followers, $p_{ik}$ is the frequency with which follower $i$ fired in rank $k$. The entropy $H_k$ was normalized by the entropy of a uniform distribution:

$$H_u = -\sum_i^n \frac{1}{n}\log_2\frac{1}{n}$$

A value of $H_k = 0$ indicates that the same neuron fired in the rank $k$ in all trials. Note that that this measure is especially sensitive to activations or failures of single followers early in a trial since they affect the ranks of all subsequent followers in that same trial. Due to our exploration of different firing rate levels, we often observed sequence failure (*Figure 4*) and numbers of followers (*Figure 2F*) much higher than the number of trials (100), both factors that limited our capacity to estimate order entropy accurately. Consequently, we limited our estimates to simulations with at least as many trials as followers and to trials where at least 25% of followers were present.

### Spatial center-of-mass

To visualize the spatial propagation of sequences, we calculated the center-of-mass of follower activations over time similar to the experimental data (see Figure 5A in *Hemberger et al., 2019*). We took the average follower XY location for those followers activating in a sliding window of 5 ms. The resulting X and Y traces were smoothed over using a Gaussian window of 15 ms.

### *K*-modes clustering

For each simulation, we summarized the activity of each follower as a single vector of 100 binary entries representing the presence (at least one spike) or silence of that follower in each of the trials (100 trials per simulation). We then applied *k*-modes clustering on each simulation separately. *K*-modes clustering is an unsupervised method for assigning data samples to $K$ clusters. It is similar to *k*-means clustering, but each cluster centroid is instead defined by the number of matching values between sample vectors and thus performs well for categorical (binary) data. Manual inspection suggested that followers were well clustered in groups of 5–10 neurons. Since we applied unsupervised clustering to 6000 simulations with very different numbers of followers each (see examples in *Figure 6D*), we took $K = n/6$ for each simulation, where $n$ is the total number of followers detected in that simulation. The algorithm had access only to spiking activity and not to the connectivity between followers. We used the implementation provided in the python package *k*-modes (https://pypi.org/project/kmodes).

### Graph visualization

All graphs of strong connections (*Figures 4B–D, 5C–H, and 6G*) were laid out using Graphviz dot and do not represent the spatial location of model neurons unless otherwise specified.

## Electrophysiology

All electrophysiological recordings used to fit the models were described in the original experimental study (*Hemberger et al., 2019*).

Adaptation index was computed as the ratio of inter-spike-interval between the last two spikes and the interspike interval of the first two spikes in whole-cell patch current-clamp recordings under a constant current injection for 1 s. The current value was two to five times the current value that elicited a single spike for each particular neuron.

## Data and software availability

All code used for model simulations and analysis has been deposited at https://github.com/comp-neural-circuits/tctx, (copy archived at swh:1:rev:4fcd1a195a64015e777e55f582e8df35c4913993; *Riquelme, 2022*). All experimental data have been previously published in *Hemberger et al., 2019*. Data used to fit the model parameters have been deposited in https://doi.org/10.6084/m9.figshare.19763017.v1.

## Acknowledgements

We thank all members of the Gjorgjieva lab for discussions, and Hiroshi Ito and Marion Silies for critical feedback on the manuscript. This work was supported by the Max Planck Society.

# Additional information

### Funding

| Funder | Grant reference number | Author |
|---|---|---|
| Max-Planck-Gesellschaft | | Juan Luis Riquelme<br>Mike Hemberger<br>Gilles Laurent<br>Julijana Gjorgjieva |

The funders had no role in study design, data collection, and interpretation, or the decision to submit the work for publication. Open access funding provided by Max Planck Society.

### Author contributions

Juan Luis Riquelme, Conceptualization, Data curation, Software, Formal analysis, Validation, Investigation, Visualization, Methodology, Writing - original draft, Writing – review and editing; Mike Hemberger, Resources, Data curation; Gilles Laurent, Resources, Supervision, Writing – review and editing; Julijana Gjorgjieva, Conceptualization, Resources, Supervision, Funding acquisition, Methodology, Project administration, Writing – review and editing

### Author ORCIDs

Juan Luis Riquelme http://orcid.org/0000-0003-4604-7405
Gilles Laurent http://orcid.org/0000-0002-2296-114X
Julijana Gjorgjieva http://orcid.org/0000-0001-7118-4079

### Decision letter and Author response

Decision letter https://doi.org/10.7554/eLife.79928.sa1
Author response https://doi.org/10.7554/eLife.79928.sa2

# Additional files

### Supplementary files

• MDAR checklist

## Data availability

All code used for model simulations and analysis has been deposited at https://github.com/comp-neural-circuits/tctx, (copy archived at swh:1:rev:4fcd1a195a64015e777e55f582e8df35c4913993). All experimental data have been previously published in Hemberger et al., 2019. Data used to fit the model parameters have been deposited in https://doi.org/10.6084/m9.figshare.19763017.v1.

The following dataset was generated:

| Author(s) | Year | Dataset title | Dataset URL | Database and Identifier |
|---|---|---|---|---|
| Riquelme JL | 2022 | Turtle cortex whole-cell patch-clamp recordings | https://doi.org/10.6084/m9.figshare.19763017.v1 | figshare, 10.6084/m9.figshare.19763017.v1 |

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
