## [Editor Report]

This is an important study of the role of spike timing in the turtle cortex. The authors provide compelling evidence that single spikes evoke motifs via strong connections, and that those motifs can be reliably routed by weaker connections. The work is careful and clear and makes intuitive predictions about how motifs are generated. It will be especially interesting to determine to what extent the results apply to the mammalian cortex.

---

## [Decision Letter]

**Decision letter after peer review:**

Thank you for submitting your article "Single spikes drive sequential propagation and routing of activity in a cortical network" for consideration by *eLife*. Your article has been reviewed by 3 peer reviewers, one of whom is a member of our Board of Reviewing Editors, and the evaluation has been overseen by Laura Colgin as the Senior Editor. The following individual involved in the review of your submission has agreed to reveal their identity: Abigail Morrison (Reviewer #3).

Essential revisions:

In a surprising display of unanimity, all three reviewers very much liked the paper, and all three reviewers had one main comment, which is related to the exceptionally low firing rates in the ex-vivo turtle cortex. Our question is: would you see repeated patterns if the firing rate were higher? Extrapolation of your simulations suggests not:

In the "Connectivity" section, starting on line 688, you ask the question "could strong connections underlie cortical sequences more generally, and are sequences an ancient feature of cortical computation?" You don't exactly answer that, but doesn't Figure 2F, which shows that the number of followers falls off rapidly with firing rates, tell us that the answer is likely to be no?

We hesitate to ask for more simulations, but without them, we're not sure this work can't be extrapolated beyond the ex-vivo turtle prep, as you appear to have done:

Line 212: "Our simulations further predict that sequences can occur under in vivo levels (high firing rates) of spontaneous activity".

Lines 594-5: "Our model produces reliable sequences from single spikes as experimentally measured ex vivo and predicts their existence in vivo, where firing rates are higher, with marked differences between excitatory and inhibitory neurons."

We looked at your Ref. 30, and Figure 8C of that reference shows firing rates in an awake turtle prep ranging from 0 and 5 spikes/s following a visual stimulus, with a mean of about 2.5 (and a max of 20). This is much higher than the 0.02-0.09 spikes/s you used in your high firing rate simulations (from line 191). We might be misreading Ref. 30, but certainly, your firing rates are 1-2 orders of magnitude lower than what's found in the mammalian cortex.

Along the same lines, on lines 468-73 you say: "In sum, we identified gate neurons in different sub-networks whose activation is critical for the activation of the sub-network. External single spikes can control the state of gate neurons and thus halt or facilitate the activation of individual sub-networks. The effect of these spikes depends on their timing relative to the sequence trigger. Finally, the activation of a sub-network may influence other sub-networks via recurrent connectivity leading to complex and non-reciprocal interactions."

We're not sure this result will be robust in a noisier network at higher rates; that should be clear.

On a secondary note, which relates to clarity, we will admit that there was somewhat of a split. Reviewer 2 found the paper "exceptionally clear and well-written", whereas Reviewer 1 found the paper often confusing. Here is a summary of what Reviewer 1 said:

My number one rule of writing is: that the reader should never have to ask "why am I being told this?". However, I had to ask that question a lot while I was reading the paper. That's because I wasn't told what to expect up front. Instead, the style was "here's an observation; here's what it means". This sounds good on paper, but it means I have to absorb all sorts of information in isolation, and then put it together. There's a simple fix to that: give us the whole story at the beginning. We do get a pretty good summary, but not until lines 316-9:

"In summary, our model suggests that rare but strong and common but weak connections play different roles in the propagation of activity: the former promotes reliable responses to single spikes, while the latter amplify spontaneous network activity and drive recurrent inhibition, effectively modulating the reliability of those responses."

It would help a huge amount to get a summary like that -- but possibly expanded, because that's only part of the story – upfront. That way, for each of the various manipulations you made -- and there were a lot of them! -- it would have been easy to tell why. As it was, for most of them I had a hard time figuring that out, and I ended up getting pretty lost.

For what it's worth, my take on your results is as follows. As you point out on lines 102-4 "each model excitatory neuron connects to other neurons with a majority of weak synapses and about two connections sufficiently strong to bring the postsynaptic neuron from rest to near firing threshold". This implies a rapid growth in the number of spikes (by a factor of 2 per relevant timestep), with saturation caused by inhibition. This is an intrinsic feature of E-I networks (London et al. (your ref 2) showed this in rat barrel cortex). But there are two differences in the turtle cortex relative to the mammalian cortex: 1) about two connections from each excitatory neuron are strong enough so that a single presynaptic spike can trigger a postsynaptic spike, and 2) the firing rate was very low (about 0.1 Hz). The low firing rate is important; as it increased, repeatability dropped, especially for inhibitory neurons (Figure 2F). So maybe we should think of the weak connectivity as controlling the level of noise.

I'm not 100% this is correct, but it is what I extracted!

Of course, my rule isn't everybody's rule, so the extent to which you implement this is up to you. But in my view, the easier a paper is to read, the more impact it has.

Besides that, we have lots of comments, mainly about clarity and figures. They're more or less collated from the three reviewers, so they're not necessarily in a totally sensible order, although we did try. To reduce the trend toward longer and longer replies to reviewers, you do not need to reply to all of these. We'll leave it to your judgment which ones you do reply to. Maybe just the substantive ones you disagree with? They're all pretty minor, so as far as we're concerned you can just implement them (or at least the ones you agree with), and we'll be happy.

1. In Figure 1A, you should be clear about what the percentages mean. We _think_ they refer to connection probabilities, but that's not mentioned in the figure caption. Under that assumption, the connectivity (per neuron) is as follows:

E-E: 93,000 x 0.14 = 13,020

E-I: 7,000 x 0.46 = 3,220

I-E: 93,000 x 0.49 = 45,570

I-I: 7,000 x 0.26 = 1,820

These connectivity numbers should be in the paper since they're relevant. And 45,570 connections per neuron seems high -- is that consistent with experiments? Or are we doing something wrong?

2. Figure 1D: a log scale would, we think, make the figure easier to read.

3. Figure 1H and I: we're guessing blue and red are E and I, respectively (since that was the color coding for the triangles in panel A). But to avoid any possibility of confusion, this should be stated in the figure caption (and/or on the plots).

4. Line 168-9: "In each simulation, we caused the trigger neuron to fire 100 action potentials at long, regular intervals". We suggest that you tell us the interval (which, from Methods, was 400 ms). We wondered what "long, regular intervals" meant when we read that.

5. You use a Poisson process as a null model. Since E-E networks tend to oscillate, we believe (although we're not 100% sure), that a Poisson process will underestimate variability. You might want to address this (assuming it's relevant, which we're not sure it is).

6. In Figure 2C, what's the y-axis? Do slight displacements indicate different spikes? This should be clear.

7. It would be really nice to show spike rasters after a trigger spike. Presumably, they won't show much of anything, but it would be important to know that -- it's possible that one can see a slight increase in firing rate.

8. Please define normalized entropy (line 177) in the main text, at least qualitatively. And in Methods, it should be defined quantitatively (currently the reader is referred to a paper).

9. We don't understand the left panel in Figure 2E.

10. Lines 257-60: "Interestingly, excitatory-to-inhibitory spike transfers are consistently shorter than their excitatory-to-excitatory counterparts, even at higher firing rates (Figure 3B inset), possibly reflecting the more depolarized state of inhibitory neurons (Figure 1H right)."

Did you mean Figure 1I, not Figure 1H?

11. It would be nice to expand on the implications of Figure 3E. If we understand things correctly (a big if), you're showing the connections between any two neurons in which the postsynaptic one fired within 100 ms of the presynaptic one. Is that correct? If so, it means there are a lot of random coincidences -- that could even happen for un-connected neurons. Which makes it hard for us to interpret what the plot means.

12. line 475: "(network and colors as in Figure 4A)". We couldn't figure out all the colors from Figure 4A.

13. Figures 6B and 6D seem inconsistent: Figure 6B shows that a&b together generate very few followers, whereas Figure 6D shows that a&b together generate a lot of followers. What are we missing?

14. lines 636-8: "The response to simultaneous activation of multiple triggers in our model suggests that sequences operate under excitatory/inhibitory balance, where local inhibition cancels excitatory signals (5) (Figure 6D, H)."

Why do Figures 6D and 6H imply that sentence? More explanation would be helpful.

15. In Methods, please include equations. At least somebody (including one of us) would want to know what they are, and the reader shouldn't have to reproduce them from the explanations (especially since the definition of the exponential integrate and fire neuron probably isn't standard). Especially important is the average number of connections/neurons.

16. In table 1 it says that the synaptic conductance time constant is 1.103681 ms. Is it really that short? That seems strange, given the long membrane time constant. And why so many significant figures?

17. The result that strong sparse connections support sequential network activity is not particularly surprising. Perhaps more emphasis could be placed on the result that the weak, dense part of the network is necessary for supporting flexible sequential activation. Could a figure be added that shows the ideal ratio or range of ratios of strong to weak connections? Perhaps this is in the results already and just needs to be brought to the foreground.

18. The matching to the data from the turtle cortex is an excellent foundation for this work and grounds the model. Still, it might be nice to know how the results change as these parameters move around. In particular, it would be nice to know which aspects of the turtle's visual cortical network and cellular properties are necessary and sufficient parameters for the observed reliable sequence generation. All this is optional, so use your best judgment as to whether you want to include any of it. And some of it might simply be a point for the Results or Discussion.

a. If the ratio of E/I in the network is changed, do the results hold?

b. Are there other parameter changes that substantively affect the results? It seems like single-cell adaptation could be an important factor in pattern generation in the network.

c. The connection pattern in the turtle cortex might also be critical for these results. This isn't modeled explicitly in this work, but it could be an important feature for the experimental results that ground the paper. Does turtle cortex "look like" rodent cortex, or are some of the connections sparser, or do they fall off spatially faster or slower? It would be good to see a discussion of this in the paper.

d. In the model, connections are random but have a log-normal distribution. How does changing this affect the results and how well does that fit rodent cortical data?

19. The Results/Discussion seem to be missing a stronger take on what this kind of patterned activity might be useful for. A concrete example of a downstream computation that relies on the reliable combinatorial activation of sub-network sequences would be useful.

20. The gating results seem very exciting! It might be nice to highlight connections to a recent explosion in ML papers on gating and putting that part of the work more front and center in the abstract.

21. Line 183: "the number of excitatory followers far exceeds the number of inhibitory followers". This seems obvious in a network with 93% excitatory neurons. Are we missing something?

22. Line 187: Hz is a unit for oscillations and spiking activity is not oscillatory. Might we suggest spikes/s?

23. Figure 2E: The MEA square is an excellent example of the sort of thing that this reviewer cannot see when printed at the intended size. Please have mercy on older eyes.

24. Line 250-254: We struggled a bit to understand the concept of spike transfers. It seems like this is any link in any sequence where the presynaptic neuron is excitatory. Could you expand this explanation, please?

25. Line 265: "very few motifs lead to the activation of excitatory neurons". This confused us because in Line 183 it says "the number of excitatory followers far exceeds the number of inhibitory followers". We are clearly missing something important; please clarify.

26. Line 305: How strong do the connections have to be? We're guessing strong enough so a single presynaptic spike causes a postsynaptic spike, but this should be stated.

27. Line 316-319: We feel that the role for strong connections is more clearly demonstrated than the role for weak connections. Obviously, if there are only a very few strong connections then the neurons will have a lower mean membrane potential and will be less responsive to spiking input. Is it that simple, or is there more to it?

28. Figure 3H: We lack the visual acuity to interpret this plot. Perhaps an alternative representation would be clearer?

29. Line 366: two "due to"s in this sentence.

30. Figure 4E-G: there are numbers on these plots that we definitely can't read, and the connectivity in 4G is not only tiny but also a very pale grey in places.

31. Figure 4E, F: this is a measured frequency rather than a probability, right?

32. Line 618: landmark -> hallmark.

33. Line 651: it seems to us that your sequence structures have got a lot in common with those reported in your ref 34 (Polychronous groups, Izhikevich), which also depends on very strong synapses. Can you expand this section with some comparison to that study?

34. Line 731: "may not be linked to behavior". Well, it's ex vivo, so we guess certainly not linked to behavior. Perhaps you can rephrase to make your meaning clearer?

35. Line 818: "variable number of connections". It would be nice to know the statistics of this and how it compares with biology

36. Line 822: Nest -> NEST. (And we appreciate you stating and citing the version used!)

---

## [Author Response]

Essential revisions:In a surprising display of unanimity, all three reviewers very much liked the paper, and all three reviewers had one main comment, which is related to the exceptionally low firing rates in the ex-vivo turtle cortex. Our question is: would you see repeated patterns if the firing rate were higher? Extrapolation of your simulations suggests not:In the "Connectivity" section, starting on line 688, you ask the question "could strong connections underlie cortical sequences more generally, and are sequences an ancient feature of cortical computation?" You don't exactly answer that, but doesn't Figure 2F, which shows that the number of followers falls off rapidly with firing rates, tell us that the answer is likely to be no?We hesitate to ask for more simulations, but without them, we're not sure this work can't be extrapolated beyond the ex-vivo turtle prep, as you appear to have done:Line 212: "Our simulations further predict that sequences can occur under in vivo levels (high firing rates) of spontaneous activity".Lines 594-5: "Our model produces reliable sequences from single spikes as experimentally measured ex vivo and predicts their existence in vivo, where firing rates are higher, with marked differences between excitatory and inhibitory neurons."We looked at your Ref. 30, and Figure 8C of that reference shows firing rates in an awake turtle prep ranging from 0 and 5 spikes/s following a visual stimulus, with a mean of about 2.5 (and a max of 20). This is much higher than the 0.02-0.09 spikes/s you used in your high firing rate simulations (from line 191). We might be misreading Ref. 30, but certainly, your firing rates are 1-2 orders of magnitude lower than what's found in the mammalian cortex.Along the same lines, on lines 468-73 you say: "In sum, we identified gate neurons in different sub-networks whose activation is critical for the activation of the sub-network. External single spikes can control the state of gate neurons and thus halt or facilitate the activation of individual sub-networks. The effect of these spikes depends on their timing relative to the sequence trigger. Finally, the activation of a sub-network may influence other sub-networks via recurrent connectivity leading to complex and non-reciprocal interactions."We're not sure this result will be robust in a noisier network at higher rates; that should be clear.

We address the reviewers' questions separately: first, whether simulations based on ex vivo turtle measurements can make predictions about in vivo turtle activity; second, whether these predictions can inform our understanding of the in vivo mammalian cortex.

In vivo turtle firing rates

We have re-analyzed some of our simulated networks and provide further details on their firing rates in a new Figure 2 —figure supplement 1, which we include and describe below and in our Results section (lines 181-200). Here we clarify the type of experimental data that guided our modeling, the activity our model produces, the limitations of experimental estimates, and the predictions of our model.

First, we clarify the experimental context for the ranges of firing rates generated in our simulations. The estimates of the mean firing rate for the in vivo turtle cortex ([0.02, 0.09] spk/s) that we used were based on the data from Fournier et al., 2018 (our previous Ref. 30) but re-analyzed for and first reported in Hemberger et al., 2019 (Figure 2 —figure supplement 1A). We have updated our citation to direct the reader to Hemberger et al., 2019 (lines 212-213) as the correct source. For that analysis, mean firing rates were extracted from all spike-sorted units detected over multiple 2-3 hour sessions in 4 lightly anesthetized turtles. Importantly, these firing rates were estimated from periods of spontaneous activity, not from periods of visual stimulation, as in Figure 8C in Fournier et al., 2018, mentioned by the reviewers. Spontaneous activity provides an estimate of the intrinsic variability of network activity in the behaving turtle without structured input. In our manuscript, we use a model to study network responses to new, externally-triggered single spikes. Hence, we focused on capturing the mean firing rates from in vivo spontaneous activity as it provides a clear baseline of uncorrelated variability for new incoming spikes, which would occur with the onset of visual stimulation. Our model reproduces this spontaneous activity through a random current (Figure 1G-I), resulting in network activity that is fully uncorrelated to the triggered spikes. To investigate the role of network firing rates in sequence propagation, we explored a range of network mean firing rates, including those compatible with the spontaneous in vivo data estimates reported in Hemberger et al., 2019 (Figure 2 figure supplement 1A).

Second, our model captures the low average firing rates of the turtle cortex, but it also produces fluctuations of higher instantaneous firing rates (Figure 2 —figure supplement 1CD). These fluctuations are also observed in the turtle cortex, often in the form of waves (Prechtl et al., 1997; Wright and Wessel, 2017). The low averages result from the strongly adaptive network dynamics and hours-long recordings (Fournier et al., 2018; Hemberger et al., 2019). In our model, convolving spike trains with a 250 ms Gaussian window (as in Fournier et al. 2018) yields peak instantaneous firing rates on the same order of magnitude as the experiments mentioned by the reviewers (maximum firing rate per neuron, averaged across all neurons, 0.53-1.36 exc. spk/s, 1.7-3.7 inh. spk/s). Note, however, that the rates in Figure 8C of Fournier et al. 2018 are averages from repeated visual stimulation, while peak rates in our model result from random fluctuations representing spontaneous activity.

Third, since it is simulated, our cortical network model allows us to access every neuron. In contrast, experimental estimates are biased by which neurons can be accessed, often those with the highest rates. For example, Fournier et al., 2018 and Hemberger et al., 2019 used extracellular recordings where spike sorting necessarily excludes most non-responding neurons from the average, potentially yielding artificially high mean rates (Levenstein and Okun, 2022; Shoham et al., 2006). For ex vivo recordings, most recorded neurons were inhibitory due to the position of the MEA. The electrical distance of excitatory neurons to the MEA limited access to the excitatory population. This bias towards inhibitory neurons in the experimental characterization of sequences was precisely one of the motivations for our modeling effort (Introduction, lines 54-56). By contrast, in our model networks, we match the density of a 2x2 mm slab of tissue and estimate the average firing rates of all neurons (100k). We find that the excitatory population in our model displays much lower firing than the inhibitory population, with most excitatory neurons producing none-to-few spikes (Figure 2 —figure supplement 1E). As a simple test: removing neurons with less than 10 spikes increases average firing rates 10-fold (from 0.02-0.13 to 0.22-1.04 spk/s) and pushes peak rates even further (to 2.2-2.3 exc. spk/s, 2.3-3.7 inh. spk/s), suggesting we might, in fact, be studying too high rates in relation to the experimental estimates which are constrained by subsampling and spike sorting.

Finally, we do find model sequences even at high rates, and we interpret them as a prediction for turtle experiments. Indeed Figure 2F shows a quick fall in the number of followers at increased rates. Still, this number remains on the order of tens of followers, even for firing rates well above estimates of in vivo spontaneous activity. We have added an example of such a sequence in Figure 2 —figure supplement 1B (mean rate 0.13 spk/s). This is a theoretical prediction specific to the turtle cortex that is experimentally testable. We now make this more explicit in our main text (lines 230, 232-240, 511, 644):

"In summary, our biologically-constrained model produces repeatable firing sequences across groups of neurons in response to single spikes in randomly chosen excitatory neurons with properties very similar to those observed in ex vivo experiments (low firing rates) that constrained the model network. Our simulations further provide an experimentally testable prediction: that sequences may occur under in vivo levels of spontaneous firing rate in the turtle cortex. In these conditions of higher spontaneous activity, the activation sequences are mainly composed of excitatory followers, whereas inhibitory followers produce less reliable and temporally jittered responses."

“Our model produces reliable sequences from single spikes as experimentally measured ex vivo and predicts their existence in vivo in the turtle cortex, where firing rates are higher, with marked differences between excitatory and inhibitory neurons.”

“Consequently, we predict that followers should be detectable in experiments in vivo in the turtle cortex but mainly within the pyramidal layer.”

“In sum, we identified gate neurons in different sub-networks whose activation is critical for the activation of the sub-network in a model based on the turtle cortex. External single spikes can control the state of gate neurons and thus halt or facilitate the activation of individual sub-networks. The effect of these spikes depends on their timing relative to the sequence trigger. Finally, the activation of a sub-network may influence other sub-networks via recurrent connectivity leading to complex and non-reciprocal interactions.”

In summary, our model operates within an electrophysiologically relevant regime of spontaneous activity in the turtle cortex and can provide an experimentally testable prediction about the presence of spiking sequences in vivo. In addition to the edits above, we include a short description of the firing rates of our model in a new subsection of our Methods section (lines 956-973):

"Simulation firing rates

We focused on capturing the mean firing rates from *ex* and in vivo spontaneous activity ([0, 0.05] and [0.02, 0.09] spk/s, Hemberger et al., 2019) because they provide a clear baseline of uncorrelated variability for new incoming spikes. To compute the mean firing rate of each simulation, we took the average during the baseline periods (100 ms before each spike injection). Our averages include all neurons (excitatory and inhibitory) equivalent to a 2x2 mm slab of tissue (Figure 1A). Since experimental estimates are limited to those high-rate neurons that can be accessed and spike-sorted in extracellular recordings (Fournier et al., 2018; Hemberger et al., 2019), comparing them to our model is not trivial. Along those lines, most excitatory neurons in our model produced none-to-few spikes (Figure 2 —figure supplement 1E), and excluding them increases mean rates 10-fold (to 0.22-1.04 spk/s).

As well as low average firing rates, our model also produces fluctuations of higher instantaneous firing rates reminiscent of the waves observed experimentally (Prechtl et al., 1997; Shew et al., 2015, Figure 2 —figure supplement 1CD). Convolving spike trains in our simulations with a 250 ms Gaussian window yields peak instantaneous firing rates one order of magnitude higher than their average (maximum firing rate per neuron, averaged across all neurons, 0.53-1.36 exc. spk/s, 1.7-3.7 inh. spk/s), and comparable to those observed in vivo under visual stimulation (Fournier et al., 2018).”

In vivo mammalian cortex

Next, we investigated average firing rates for mammalian cortices and clarified the limits of extrapolating our results to mammals.

First, while certainly lower than in mammals, spontaneous firing rates in the turtle cortex might not be as different as the reviewers propose. Architectonically and evolutionarily, the turtle cortex is most comparable to mammalian 3-layered paleo- and archi-cortices (piriform and hippocampal formation, Fournier et al., 2015). In hippocampus CA1, firing rates of pyramidal cells are known to span from 0.001 to 10 spk/s and follow heavily skewed distributions, with >70% of neurons showing mean spontaneous rates of <1 spk/s, even during active running (Mizuseki and Buzsáki, 2013). A recent re-analysis of data from several regions of the rodent forebrain suggests that neurons display a “ground state” firing mode characterized by irregular spiking at very low rates, with CA1 and piriform cortex often showing 0.1-1 spk/s (see Figures 2C and 4D of Levenstein et al., 2021).

Low spontaneous firing rates (<1 spk/s) have also been reported across neocortical areas (see reviews by Barth and Poulet, 2012; Shoham et al., 2006), sparking interest in theories of sparse-firing operation of cortex (Wolfe et al., 2010). A recent estimate of firing rates based on multi-area neuropixel data suggests that 40% of the neurons across a column of in vivo mouse sensory cortex fire at <1 spk/s (Levenstein and Okun, 2022). Finally, all the caveats regarding mean firing rate estimations described above for turtle studies equally apply to mammalian literature, including differences between evoked vs. spontaneous activity and the effects of biased sampling that result from short recordings, electrode location, and spike sorting.

Second, our manuscript only makes predictions for the turtle cortex at the spontaneous firing rate regimes that have been experimentally reported for it. We discuss similarities with other cortices in the hope that our work may guide future research in those systems. Indeed, long-tailed distributions of strong connections and spiking sequences, as observed in the turtle cortex and our model, have also been reported for in vivo mammalian cortices, from mice to humans (Discussion, lines 756-768). However, our model is tightly constrained by turtle measurements, as we now clearly state in the manuscript. Making concrete quantitative predictions about sequences in the mammalian cortex would require different species-specific models, which are beyond the scope or intention of our manuscript. In addition to the text changes described in the previous section, we now address this explicitly in our discussion (lines 768-776):

“Thus, could strong connections underlie cortical sequences more generally, and are sequences an ancient feature of cortical computation? We based our model on data from the turtle cortex, yet other species may introduce variations that might be relevant, including differences in operating firing rates or neuronal populations. For instance, the reliability of sequence propagation may depend on connectivity features we did not explore here, such as the presence of different connectivity motifs or differences in the distance-, layer- and population-specific connection probabilities (Jiang et al., 2015; Song et al., 2005). Further modeling and experimental work on the cortical responses of diverse species will be needed to provide a comprehensive comparative perspective (Laurent, 2020).”

Sequences at spontaneous rates beyond turtle in vivo

Nonetheless, we have taken this opportunity to explore sequence propagation in simulations at higher rates. We describe these in Author response image 1 (below). This initial exploration suggests that the mechanism we describe applies under spontaneous rates much higher than the in vivo turtle estimates. Still, it also brings up limitations for follower detection that need to be addressed in that scenario, which is beyond the scope of our manuscript.

We re-run the example simulations from Figure 2 —figure supplement 1 under increasingly higher mean currents (200-2000 pA, Author response image 1). We found that mean inputs higher than these caused synchronous firing while increasing the input variance did not strongly affect the firing rates. The resulting mean rate reached up to 2.28 spk/s, with inhibitory neurons firing at rates several orders of magnitude higher than excitatory neurons (Author response image 1, mean rate: 0.36 exc spk/s, 27.75 inh spk/s; mean peak rate: 2.51 exc. spk/s, 33.71 inh. spk/s). The rates of excitatory neurons follow a high-variance, skewed distribution which is no longer well captured by our Poisson null model (Author response image 1; compare to Figure 2 —figure supplement 1E) and which results in a wide distribution of firing rate modulation (∆FR, difference of firing rate before and after spike induction, Author response image 1; compare to Figure 2B). In consequence, our statistical test could no longer differentiate followers and non-followers. Therefore, to observe sequence propagation in this extended firing rate regime, we tracked the neurons identified as followers in our original simulations (Author response image 1). Followers continued to activate following the induced spike, even under the strongest drive but did so very sparsely (average follower activated in 15% of trials). Consequently, the original sequences never repeated in full but presented, across trials, the same type of sub-sequence failures we studied in Figures 4-6. These failures are consistent with our original analysis of recurrent interactions, where we estimated that spikes from 90% of the neurons in the network would produce lateral inhibition onto a given subsequence (Figure 6H). Given the large number of ongoing spontaneous excitatory spikes in the network (around 3,400 spikes in the 100 ms surrounding any induced spike), it is not surprising that individual subsequences should be frequently halted.

Our model suggests that finding sequences at high firing rates would be greatly aided by detecting followers at lower firing rates. This suggestion is relevant to study sequences in experimental setups. For instance, the low spontaneous rates and strong adaptive features of the turtle cortex may make it especially suited to the study of spiking sequences. Similarly, in a mammalian network, repeating sequences can be detected in the rat somatosensory cortex in vivo during the transition from low-firing (DOWN) to high-firing (UP) states but not within the high-firing states (Luczak et al., 2007).

**Author response image 1. sa2fig1:** Sequence evolution at super-high firing rates. (A) Input-output curve of the network, including new simulations at higher firing rates. Original simulations explored the area in purple (same as Figure 2 —figure supplement 1). New simulations as colored dots. Input standard deviation for new simulations: 110 pA. (B) Spike raster of a simulation with 2.28 spk/s mean rate (red dot in A). All spikes for all neurons in the network for 1 second in the middle of the simulation. Arrowheads indicate the initiation of a trial. Purple circles highlight injected spikes. Blue: exc.; pink: inh. (C) Distribution of single-cell mean firing rate for an example simulation with mean network rate of 2.28 spk/s (red dot in A, mean exc. rate of 0.36 spk/s). Only excitatory neurons are shown. Dashed line indicates a Poisson fit. (D) Evolution of sequences as firing rates increase above turtle in vivo estimates. Each row corresponds to one of the three original simulations in Figure 2 —figure supplement 1 (s1, s2, s3). Left column: example trials from original simulations. Other columns: example trials of each simulation re-run at an increased mean firing rate (indicated at the top; colored dots as in A). Only spikes from excitatory followers are shown. Purple: trigger spike. Within each row, we show spikes from the same neurons with the same sorting. (E) Distribution of single-cell firing rate modulation (∆FR) for an example simulation with 2.28 spk/s mean rate (red dot in A). Yellow: ∆FR for excitatory followers as detected in original simulations. Blue: other excitatory neurons in the network.

On a secondary note, which relates to clarity, we will admit that there was somewhat of a split. Reviewer 2 found the paper "exceptionally clear and well-written", whereas Reviewer 1 found the paper often confusing. Here is a summary of what Reviewer 1 said:My number one rule of writing is: that the reader should never have to ask "why am I being told this?". However, I had to ask that question a lot while I was reading the paper. That's because I wasn't told what to expect up front. Instead, the style was "here's an observation; here's what it means". This sounds good on paper, but it means I have to absorb all sorts of information in isolation, and then put it together. There's a simple fix to that: give us the whole story at the beginning. We do get a pretty good summary, but not until lines 316-9:"In summary, our model suggests that rare but strong and common but weak connections play different roles in the propagation of activity: the former promotes reliable responses to single spikes, while the latter amplify spontaneous network activity and drive recurrent inhibition, effectively modulating the reliability of those responses."It would help a huge amount to get a summary like that -- but possibly expanded, because that's only part of the story – upfront. That way, for each of the various manipulations you made -- and there were a lot of them! -- it would have been easy to tell why. As it was, for most of them I had a hard time figuring that out, and I ended up getting pretty lost.For what it's worth, my take on your results is as follows. As you point out on lines 102-4 "each model excitatory neuron connects to other neurons with a majority of weak synapses and about two connections sufficiently strong to bring the postsynaptic neuron from rest to near firing threshold". This implies a rapid growth in the number of spikes (by a factor of 2 per relevant timestep), with saturation caused by inhibition. This is an intrinsic feature of E-I networks (London et al. (your ref 2) showed this in rat barrel cortex). But there are two differences in the turtle cortex relative to the mammalian cortex: 1) about two connections from each excitatory neuron are strong enough so that a single presynaptic spike can trigger a postsynaptic spike, and 2) the firing rate was very low (about 0.1 Hz). The low firing rate is important; as it increased, repeatability dropped, especially for inhibitory neurons (Figure 2F). So maybe we should think of the weak connectivity as controlling the level of noise.I'm not 100% this is correct, but it is what I extracted!Of course, my rule isn't everybody's rule, so the extent to which you implement this is up to you. But in my view, the easier a paper is to read, the more impact it has.

We thank the reviewers for their constructive suggestions to improve the clarity and impact of our manuscript. We have added short summary sentences at the beginning of each of our result sections to guide the reader, as suggested by Reviewer 2 (lines 278-280, 378-381, 454-458, 543-546):

“To better understand the mechanisms behind follower activation, we examined how spikes propagate through our model networks. We found that single strong sparse connections drive reliable excitatory responses, while convergent weak connections control the level of spontaneous network activity and drive inhibition.”

“To better understand the regulation of sequential neuronal activations, we examined when and how sequences fail to propagate. We found that sequences could fail partially, with different sections of the same sequence failing to propagate independently of one another.”

“We hypothesized that the connections between sub-networks could be critically important for sequence propagation. Indeed, the excitability and state of an entry point to a sub-network should affect sub-network activation, hence, we refer to these entry point neurons as “gates”. By injecting a single extra spike to these gates, we could control the activation of sub-sequences and thus the path of propagation of the sequence.”

“To examine how sequences interact, we studied a new set of simulations where multiple trigger neurons were activated simultaneously. We found that sequences from coactivated trigger neurons often interact, but they do so reliably, resulting in a combinatorial definition of followers.”

The take on our results by Reviewer 2, although mainly correct, lacks a key element: in our model, unlike in London et al., 2010, the network neurons showing a “rapid growth in spikes” in response to an induced spike are always the same ones (which we call followers). Thus, the growth of activity is channeled within the network and results in repeatable sequences of activations that can be reliably modified (which we interpret as routing).

Reviewer 2 also states that the number of strong connections is an important difference between the turtle and mammalian cortex. Long-tailed distributions of synaptic strengths are reported for mammalian cortices (Discussion, lines 756-768), but, to our knowledge, the exact number of strong connections per neuron is unknown. Even the definition of “strong” will vary when considering differences in network and single-cell properties (for example, adaptation currents, resting and threshold potentials). Thus, it’s hard to establish how much of a difference there is between mammalian cortices and our model’s estimate for the turtle dorsal cortex.

Besides that, we have lots of comments, mainly about clarity and figures. They're more or less collated from the three reviewers, so they're not necessarily in a totally sensible order, although we did try. To reduce the trend toward longer and longer replies to reviewers, you do not need to reply to all of these. We'll leave it to your judgment which ones you do reply to. Maybe just the substantive ones you disagree with? They're all pretty minor, so as far as we're concerned you can just implement them (or at least the ones you agree with), and we'll be happy.1. In Figure 1A, you should be clear about what the percentages mean. We _think_ they refer to connection probabilities, but that's not mentioned in the figure caption. Under that assumption, the connectivity (per neuron) is as follows:E-E: 93,000 x 0.14 = 13,020E-I: 7,000 x 0.46 = 3,220I-E: 93,000 x 0.49 = 45,570I-I: 7,000 x 0.26 = 1,820These connectivity numbers should be in the paper since they're relevant. And 45,570 connections per neuron seems high -- is that consistent with experiments? Or are we doing something wrong?

The percentages are estimated local connection probabilities. Our connection probabilities are distance-dependent with a Gaussian decay profile (given in Figure 1C). From our main text:

“Because our estimates of connection probabilities (Figure 1A) are derived from paired-patch recordings of nearby neurons, we scaled the decay profile to match population-specific probabilities in any given disc of 200μm radius.”

Note that our connectivity degrees are not fixed. We updated our Methods section to make this more explicit (lines 920-924):

“The degrees of connectivity were not fixed, so inhomogeneities in neuron location and randomness of connectivity meant that neurons received a variable number of connections.

For instance, the distributions of numbers of E-E weak and strong connections are given in Figure 1E.”

We have adjusted our schematic Figure 1A to highlight the locality of our connectivity and changed its caption to include the average number of connections:

2. Figure 1D: a log scale would, we think, make the figure easier to read.

We believe the linear scale shows more clearly the difference between the very strong and sparse connections at the far end of the tail (above orange arrowhead) compared to the weaker and dense connections at the main body (below blue arrowhead). We add a log-scale version in Author response image 2.

**Author response image 2. sa2fig2:** Log-scale distribution of connection strengths. Top: histogram of conductances for all excitatory-to-excitatory connections in an example model network. Bottom: histogram of synaptic conductances estimated from EPSP amplitudes experimentally obtained from recorded pairs of connected neurons.

3. Figure 1H and I: we're guessing blue and red are E and I, respectively (since that was the color coding for the triangles in panel A). But to avoid any possibility of confusion, this should be stated in the figure caption (and/or on the plots).

The reviewers are correct. We have added this to Figure 1H-I captions.

4. Line 168-9: "In each simulation, we caused the trigger neuron to fire 100 action potentials at long, regular intervals". We suggest that you tell us the interval (which, from Methods, was 400 ms). We wondered what "long, regular intervals" meant when we read that.

We have updated this line to include the 400 ms period.

5. You use a Poisson process as a null model. Since E-E networks tend to oscillate, we believe (although we're not 100% sure), that a Poisson process will underestimate variability. You might want to address this (assuming it's relevant, which we're not sure it is).

Within the spontaneous firing rate regimes that we consider, we find the Poisson captures well the variance and shape of single-cell firing rates, even at very high rates (see new Figure 2 —figure supplement 1E). The reviewers are correct that a Poisson approximation fails to capture the variance at much higher firing rates than those observed in the turtle cortex (Author response image 1). We refer the reviewers to our response (above) to their main comment about mammalian firing rates.

6. In Figure 2C, what's the y-axis? Do slight displacements indicate different spikes? This should be clear.

Vertical displacement indicates different follower neurons. We have labeled the y-axis in Figure 2C and specified this in the caption.

7. It would be really nice to show spike rasters after a trigger spike. Presumably, they won't show much of anything, but it would be important to know that -- it's possible that one can see a slight increase in firing rate.

We now show spike rasters including all neurons for the two simulations shown in Figure 2C in our new Figure 2 —figure supplement 1, which we include and describe in our response to the reviewers’ main comment about firing rates (above). The average rates in our simulations are low, but given the size of our network (100k neurons), they translate into thousands to millions of spikes per simulation. Note that there are fluctuations of network activity, but these are not aligned with the injected spikes in our rasters (Figure 2 —figure supplement 1C) but rather reflect random spontaneous activity.

The reviewers are correct that there is, on average, an increase in the firing rate as a consequence of the injected spike, but this increase is not visible to the eye in the spike rasters. We measure the increase of single-cell firing rate (∆FR) to detect followers and find that it is centered around zero for the majority of the network (Figure 2B). The average ∆FR across all neurons (including followers) for all our simulations is between 0.003-0.005 spk/s.

8. Please define normalized entropy (line 177) in the main text, at least qualitatively. And in Methods, it should be defined quantitatively (currently the reader is referred to a paper).

We have updated this line in the main text to read (lines 194-198):

“When ordering the followers in the model network by activation delay from the trigger neuron spike, we observed reliable spike sequences as seen in experiments (Figure 2C). We used an entropy-based measure to quantify the variability of follower identity per rank in a sequence and observed similar results as in the experiments, with follower ordering being most predictable in the first four ranks of the sequence (see Methods, Figure 2D).” And in our methods (lines 1018-1031):

“To compare the reliability of the order of follower activation to that seen in experiments, we used the same entropy-based measure as used first in experiments (5) (Figure 2D). For every trial, followers were ordered by the time of their first spike, and we calculated the entropy of follower identity for each rank:Hk=−∑inpiklog2⁡pik where n is the number of followers, *p_ik_* is the frequency with which follower *i* fired in rank *k*. The entropy *H_k_* was normalized by the entropy of a uniform distribution:Hu=−∑in1nlog2⁡1nA value of *H_k_* = 0 indicates that the same neuron fired in the rank *k* in all trials. Note that this measure is especially sensitive to activations or failures of single followers early in a trial since they affect the ranks of all subsequent followers in that same trial. Due to our exploration of different firing rate levels, we often observed sequence failure (Figure 4) and numbers of followers (Figure 2F) much higher than the number of trials (100), both factors that limited our capacity to estimate order entropy accurately. Consequently, we limited our estimates to simulations with at least as many trials as followers and to trials where at least 25% of followers were present.

9. We don't understand the left panel in Figure 2E.

We reproduced the analysis of the center-of-mass of follower activations as was observed experimentally (Hemberger et al., 2019) Figure 5A. It indicates that follower activations spread away from the source neuron.

For clarity, we have updated our Figure 2E and its caption:

“Figure 2E. Left: Spatial evolution of the center-of-mass of follower activations during the first 100 ms of the first sequence in C. Trigger neuron in purple outline. Exc followers in blue outlines.”

And added to our Methods (lines 1032-1037):

**“**Spatial center of mass

To visualize the spatial propagation of sequences, we calculated the center-of-mass of follower activations over time similar to the experimental data (see Figure 5A in Hemberger et al. 2019). We took the average follower XY location for those followers activating in a sliding window of 5 ms. The resulting X and Y traces were smoothed over using a Gaussian window of 15 ms.”

10. Lines 257-60: "Interestingly, excitatory-to-inhibitory spike transfers are consistently shorter than their excitatory-to-excitatory counterparts, even at higher firing rates (Figure 3B inset), possibly reflecting the more depolarized state of inhibitory neurons (Figure 1H right)."Did you mean Figure 1I, not Figure 1H?

The reviewer is correct. Apologies. This was a typo which we have now corrected.

11. It would be nice to expand on the implications of Figure 3E. If we understand things correctly (a big if), you're showing the connections between any two neurons in which the postsynaptic one fired within 100 ms of the presynaptic one. Is that correct? If so, it means there are a lot of random coincidences -- that could even happen for un-connected neurons. Which makes it hard for us to interpret what the plot means.

The reviewers understand Figure 3E correctly, with the only caveat that we only consider neurons that share connections, so we don’t quantify coincidences of un-connected neurons. See also our updated description of a spike transfer in our response to reviewers’ comment 24.

Figure 3E (weights of connections involved in spike transfers) can be understood as a sampling, with replacement, from Figure 1D (weights of all connections in the network). Deviations in the shape of the distribution thus indicate connection strengths that are traversed more frequently than one would expect from purely random sampling. We now write in the main text (lines 308-315):

“We found that the strength distribution of those connections underlying spike transfers matches the shape of the full connectivity, with a peak of very weak connections followed by a long tail (as in Figure 1D). Interestingly, the distribution of excitatory-to-excitatory spike transfers displays a secondary peak indicating an over-representation of strong connections (Figure 3E top inset). By contrast, the absence of this secondary peak and the much higher primary peak (~1.5M) for excitatory-to-inhibitory spike transfers suggest that inhibitory spikes are primarily the result of weak convergence (Figure 3E bottom).”

12. line 475: "(network and colors as in Figure 4A)". We couldn't figure out all the colors from Figure 4A.

The reference should be to Figure 4B. Apologies. This was a typo which we have now corrected.

13. Figures 6B and 6D seem inconsistent: Figure 6B shows that a&b together generate very few followers, whereas Figure 6D shows that a&b together generate a lot of followers. What are we missing?

In Figure 6B, “a&b” indicates neurons that are followers to both a in isolation and to b in isolation. It shows an almost empty intersection between the set of followers to a and the set of followers to b. In Figure 6D we consider followers to the coactivation of both neurons at once, which we denote by “ab”.

To clarify this, we have added text labels to our schematics in Figure 6A and C.

Extract from Figure 6

A. Schematic of follower classes as a function of trigger neuron (purple: follower to a; red: follower to b; blue: follower to both a and b).

C. Schematic of follower classes as a function of trigger: single trigger neuron (a or b) or simultaneous coactivation (ab). Blue, purple, and red, as in A.

14. lines 636-8: "The response to simultaneous activation of multiple triggers in our model suggests that sequences operate under excitatory/inhibitory balance, where local inhibition cancels excitatory signals (5) (Figure 6D, H)."Why do Figures 6D and 6H imply that sentence? More explanation would be helpful.

We have rewritten this sentence and parts of its paragraph to (lines 690-700):

“Although a single spike can reliably trigger many others in the network, we never observed explosive amplification of activity in our model. On the contrary, multiple excitatory spikes result in a sub-linear summation of responses (Figure 6D), likely reflecting the frequent lateral inhibition onto sub-sequences (Figure 6H). Together, this suggests that sequences operate in a regime where local inhibition often balances excitatory signals. We did not explicitly construct our model to display an excitatory/inhibitory balance. Still, this form of feedback inhibition and the possibility of routing via the alteration of excitatory and inhibitory conductances in gate neurons is reminiscent of “precise balance”, a form of balance within short temporal windows (Bhatia et al., 2019). Hence, our work is consistent with and extends previous theoretical work proposing that the gating of firing rate signals could be arbitrated via the disruption of excitatory/inhibitory balance in local populations (Vogels and Abbott, 2009).

15. In Methods, please include equations. At least somebody (including one of us) would want to know what they are, and the reader shouldn't have to reproduce them from the explanations (especially since the definition of the exponential integrate and fire neuron probably isn't standard). Especially important is the average number of connections/neurons.

We have updated our methods to include equations for the exponential integrate-and-fire neuron.

“We simulated neurons as adaptive exponential integrate-and-fire (see Table 1 for parameters) (Brette and Gerstner, 2005). We used the NEST model “aeif_cond_exp” (see Methods, Simulations), which implements a conductance-based exponential integrate-and-fire neuron model where the membrane potential is given by:

CdVdt=−gL(V−EL)+gLΔTexp⁡V−VTΔT−ge(V−Ee)−gi(VC−Ei)−w=Ieτwdwdt=a(V−EL)−wAt each firing time the variable is increased by an amount *b*, which accounts for spike-triggered adaptation. Inhibitory and excitatory synaptic conductances (*g_i_* and *g_e_*) are implemented as exponential decay kernels (see NEST documentation, Linssen et al., 2018).”

We also include equations for the normalized rank entropy in Methods (see reviewers’ comment 8) and the number of connections in the caption of Figure 1A (see reviewers’ comment 1).

16. In table 1 it says that the synaptic conductance time constant is 1.103681 ms. Is it really that short? That seems strange, given the long membrane time constant. And why so many significant figures?

It is difficult to establish what values are “strange” since we lack alternative sources to compare turtle cortex physiology. All of our model parameters take their values either from fitting procedures or literature. The fitting of membrane and synapse time constants was “bottom-up”, that is, they were determined from single- or paired-clamp measurements, and we did not perform any parameter searches that would produce a particular network behavior. The values resulting from the fitting could certainly be rounded after a few decimals, but we report them as they were used in our simulations.

We next describe in more detail the fitting process that we used to obtain the membrane and synaptic time constants and expand our Methods sections (see updated text below). We first fitted the membrane leak conductance and the membrane capacitance (which together determine the membrane time constant) from current-clamp traces. The result of this fit, as well as example traces, are in Figure 1B. We then fitted the synaptic conductance time constant using ESPS rise times. Examples of the resulting model EPSPs are shown as an inset in Figure 1D. All data used here are publicly available through figshare (see Data and Software availability).

Extract from Figure 1

B. Top left: example of fit (black) of membrane potential responses to current injection in a recorded neuron (blue). Bottom left and right: distributions of fitted membrane capacitance (C_m_) and leak conductance (g_L_) (n=3,886 traces).

D. Lognormal fit to peak excitatory synaptic conductances. Top: cumulative distribution function (cdf). Bottom: probability density function (pdf). Conductances were estimated from EPSP amplitudes experimentally obtained from recorded pairs of connected neurons (gray dots). Inset: example of modeled EPSPs for different synaptic weights (top arrowheads) in an isolated postsynaptic model neuron at resting potential.

We expand our Methods (lines):

“To estimate membrane properties, we used least squares to match the output of our neuron model to the membrane potential of real neurons in whole-cell clamp under different levels of current injection (Hemberger et al., 2019). An example of the fit can be seen in Figure 1B. We first fit the model membrane leak conductance (resistance) using the steady state of the experimentally measured membrane potential after current injection. We then fit the model membrane capacitance (which determines the membrane time constant) using the experimentally measured potential immediately after the onset of current injection. We fitted all traces independently (n=3,886), and we used the median of the distributions for all neurons in the model.

(…)

We fitted synaptic time constants to rise times of excitatory postsynaptic potentials (EPSPs, n=382) obtained in paired patch recordings using least-squares and used the median for our model synapses. We then converted experimentally measured EPSP amplitudes (same n=382) to synaptic conductances (Figure 1D) using least squares and fitted a lognormal distribution via maximum likelihood estimation.“

17. The result that strong sparse connections support sequential network activity is not particularly surprising. Perhaps more emphasis could be placed on the result that the weak, dense part of the network is necessary for supporting flexible sequential activation. Could a figure be added that shows the ideal ratio or range of ratios of strong to weak connections? Perhaps this is in the results already and just needs to be brought to the foreground.

We respectfully disagree with the statement that reliable sequential activity is self-evident, given the presence of strong sparse connections. In fact, even the reviewers’ main comment seems to attribute our results to the “exceptionally low firing rates of the turtle cortex,” not to the presence of strong sparse connections. Our networks are large and recurrent, with parameters constrained by data and subject to uncorrelated random activity, so it is non-trivial to conclude that a single factor supports sequence generation. Both strong-sparse and weak-dense connections are important for understanding how sequences can be generated and propagated.

We do not per se have two separate groups of strong and weak connections, but rather a continuum of strengths since the entire distribution of connection strengths is extracted from experimental data (Figure 1D). We refer to them as weak and strong based on thresholds that we selected to study the two modes of the distribution of traversed connections (bottom 90% or top 0.3%, lines 328-333, Figure 3E top). Therefore, it is not obvious how we could vary the ratio of strong to weak connectivity unless we completely change our model.

Nonetheless, we next clarify how weak connections are important for sequence flexibility. While propagation happens mainly through strong connections, the actual efficacy of those strong connections will depend on the conductance state of followers, as we showed through the manipulation of gate neurons (Figure 5). Our analysis shows that weak connections play two key roles in altering this conductance state:

1. Weak connections provide an input amplification effect, depolarizing the network and promoting or reducing the number of followers. In Figure 3H (updated in reviewers’ comment 28), we directly compared simulations with and without weak connections. We observed that weak connections could differentially modulate the number of reliably-responding neurons as a function of the input regime of the network. Since activity in the turtle cortex evolves spatially as traveling waves (Figure 2 —figure supplement D) (Prechtl et al., 1997), weak connections play a unique role in enabling or disabling propagation within different subcircuits.

2. Weak connections provide the main lateral inhibition between subsequences. In our model, inhibitory neurons are widely connected and driven by weak convergent excitatory connections (Figure 3DE). When a trigger neuron spikes and activates its followers, these together will cause lateral inhibition onto downstream sub-sequences in 90% of the cases (Figure 6H). This lateral inhibition between sub-sequences, mediated through weak connections, can reliably halt propagation, creating a class of followers that require both the activation of one trigger neuron and the silence of another (which we term combination-specific followers, Figure 6E, lines 565-570).

We introduce both roles of weak connections supporting flexible routing in our discussion of tuning (as untuned depolarization and feature competition, lines 828-834, 848-856):

“In mouse V1, stimulus presentation triggers broadly-tuned depolarization via a majority of weak connections, while sparse strong connections determine orientation-selective responses (Cossell et al., 2015). This untuned depolarization via weak connections is consistent with our finding of activity regimes where weak connections can modulate the reliability of propagation (Figure 3H). Overall, the dual roles in routing played by weak and strong connectivity may have corresponding roles in what we describe as tuning.

(…)

Under this view, lateral inhibition between sub-networks, mediated through the convergence of weak connections (Figure 3ED), may perform nontrivial computations on the properties of stimuli in a sensory cortex. Indeed, lateral inhibition triggered by early responding principal neurons implements concentration-invariant coding in mouse olfactory cortex (Bolding and Franks, 2018). In line with our observation of common lateral inhibition between sub-networks (Figure 5H, Figure 6H), activations of single neurons in mouse V1 tend to inhibit neurons with overlapping tuning (feature competition) (Chettih and Harvey, 2019)”

18. The matching to the data from the turtle cortex is an excellent foundation for this work and grounds the model. Still, it might be nice to know how the results change as these parameters move around. In particular, it would be nice to know which aspects of the turtle's visual cortical network and cellular properties are necessary and sufficient parameters for the observed reliable sequence generation. All this is optional, so use your best judgment as to whether you want to include any of it. And some of it might simply be a point for the Results or Discussion.a. If the ratio of E/I in the network is changed, do the results hold?b. Are there other parameter changes that substantively affect the results? It seems like single-cell adaptation could be an important factor in pattern generation in the network.c. The connection pattern in the turtle cortex might also be critical for these results. This isn't modeled explicitly in this work, but it could be an important feature for the experimental results that ground the paper. Does turtle cortex "look like" rodent cortex, or are some of the connections sparser, or do they fall off spatially faster or slower? It would be good to see a discussion of this in the paper.d. In the model, connections are random but have a log-normal distribution. How does changing this affect the results and how well does that fit rodent cortical data?

We thank the reviewers for appreciating our efforts to ground the model. These are all interesting questions and particularly relevant for expanding our results to the mammalian cortex. We next address each suggestion individually.

Suggestion A: Ratio of E/I

Single-cell voltage clamp experiments suggest that inhibitory postsynaptic currents in the turtle cortex are 2-3 times stronger than excitatory currents. Consequently, we modeled the distribution of inhibitory synaptic strengths as a scaled-up version of the excitatory distribution, for which we had paired-patch measurements (Figure 1D, Methods). For the ratio of E/I neurons in the network, we took our best estimate of the interneuron population in the turtle dorsal cortex, which is 7% (see Hemberger et al., 2019, C.M. Müller, personal communication). Changing either the ratio of E/I strengths or E/I neurons can change the efficacy and expected number of strong excitatory connections and affect our conclusions. However, we believe that such changes should be, like our model, grounded on estimates that are species-specific. Thus, we have included in our discussion as part of our response to the reviewers’ main comment about extrapolations to mammals:

“Thus, could strong connections underlie cortical sequences more generally, and are sequences an ancient feature of cortical computation? We based our model on data from the turtle cortex, yet other species may introduce variations that might be relevant, including differences in operating firing rates or neuronal populations.”

Suggestion B: Single-cell adaptation

From whole-cell patch-clamp recordings, we estimate that the majority of excitatory neurons in the turtle cortex are adaptive, consistent with the strongly adaptive responses observed under visual stimulation (Fournier et al. 2018). On the other hand, inhibitory neurons show a diverse range in their adaptation/facilitation, likely reflecting inhibitory diversity in the turtle cortex, which is transcriptomically comparable to that in the mammalian cortex (Tosches et al. 2018). For simplicity, our model assumed all neurons, excitatory or inhibitory, to be adaptive. We have re-run a subset of our simulations where we disabled adaptive currents for all inhibitory neurons (*a* = 0, *b* = 0, see equations in reviewers’ comment 15). We provide the results in Author response image 3. Our conclusions regarding the relationship between firing rate and the number of detected followers remain unchanged (Author response image 3; compare to Figure 2F). The resulting networks still produce reliable sequences with similar properties (Author response image 3). Compared to our original simulations with adaptive inhibitory neurons, sequences without inhibitory adaptation display fewer excitatory followers (mean exc follower count without inh adaptation: 7; with: 20) and more inhibitory followers (without: 8; with: 4). Having explored these two extremes (all inhibitory neurons being adaptive or all non-adaptive), we expect that a mixed model that includes diverse inhibitory adaptation would still display sequences like the ones we present.

**Author response image 3. sa2fig3:** Sequences without inhibitory neuron adaptation. (A) Number of followers in a network with non-adaptive inhibitory neurons detected for each simulation as a function of the mean level of activity in the network. Blue: exc; red: inh; thick line: moving average; Gray dots: sequences in B. (B) Top to bottom: sequences of follower spikes in two trials for three example simulations (gray dots in A). Y-axis: Followers sorted by median spike-time. Same sorting in both trials. Spikes from non-follower spikes are not shown.

Suggestion C: Connectivity patterns

We agree that specific patterns of structured connectivity in the turtle cortex could be important for the generation and propagation of sequences. Unfortunately, we lack data on the fine-grained connectivity in the turtle cortex that would be relevant for the recurrent interactions that we study in our model. This lack of data led us to choose random connectivity for our model, although we did impose spatially decaying connection probability based on experimental estimates. Even under the assumption of randomness, we demonstrate reliable propagation, establishing a baseline against which one could evaluate different theoretical connectivity patterns (such as synfire or polychronous chains, which we describe in our Discussion, lines 702-726).

As the reviewers suggest, rodent and turtle cortical connectivity display certain spatial similarities. For instance, both the turtle dorsal cortex and the rodent auditory cortex display Gaussian profiles of distance-dependent connectivity (Levy and Reyes, 2012), and sensory afferents produce a gradient of en-passant synapses for both the turtle dorsal cortex and the rodent piriform cortex (Fournier et al., 2015). However, data is limited and detailed comparisons across the different cortical areas and species are not trivial. Nonetheless, we discuss the issue of connectivity patterns and potential future directions in our updated Discussion section:

“For instance, the reliability of sequence propagation may depend on connectivity features we did not explore here, such as the presence of different connectivity motifs or differences in the distance-, layer- and population-specific connection probabilities (Jiang et al., 2015; Song et al., 2005). Further modeling and experimental work on the cortical responses of diverse species will be needed to provide a comprehensive comparative perspective (Laurent, 2020).

(…)

Certain nonrandom features may be expected in turtle cortical connectivity, such as axonal projection biases (Shein-Idelson et al., 2017), differences between apical and basal dendritic connectivity, gradients of connectivity across the cortical surface (Fournier et al., 2015), or plasticity-derived structures. Transcriptomic and morphological evidence suggests that the turtle cortex contains diverse neuronal types, as seen in the mammalian cortex (Nenadic et al., 2003; Tosches et al., 2018). Different neuronal types may play different roles in information routing.”

Suggestion D: Log-normal distributions

In our present results, we already explored changing the log-normal distribution by truncating its tail (Figure 3F, reproduced below). Although similar in 90% of its connectivity, we found that the resulting network could rarely produce any followers.

Extract from Figure 3

“F. Top: schematic of alternative network models and their truncated distribution of synaptic strengths. Bottom: Number of detected excitatory followers per simulation (n=2,000 each). Boxes: median and [25th, 75th] percentiles.”

Additionally, the reviewers ask how well a log-normal fits rodent cortical data. Log-normal distributions of synaptic strengths are common to many mammalian species, and we include a short summary in our Discussion (below, lines 756-768). However, the specific lengths and thicknesses of the tails vary, and we cannot interpret them without the context of the particular circuit’s neuronal properties.

“Such sparse and strong connections are a common feature of brain connectivity. Indeed, long-tailed distributions of connection strengths are found in rat visual cortex (Song et al., 2005), mouse barrel and visual cortices (Cossell et al., 2015; Lefort et al., 2009), rat and guinea pig hippocampus (Ikegaya et al., 2013; Sayer et al., 1990), and human cortex (Shapson-Coe et al., 2021).”

19. The Results/Discussion seem to be missing a stronger take on what this kind of patterned activity might be useful for. A concrete example of a downstream computation that relies on the reliable combinatorial activation of sub-network sequences would be useful.

We agree this is a very exciting question, albeit difficult to answer with our current model based on ex vivo activity in the turtle cortex. Our study proposes sequential propagation (or failure to propagate) as an indication of activity routing. This routing effectively produces reliable mappings from specific initial inputs (injected spikes) to specific activations (followers), i. e. a computation. Given that we study a visual cortex, we discuss the reliable routing of spikes as implementing visual tuning of cortical neurons, one of the core cortical computations underlying perception. We now write this more explicitly in our Discussion (below, lines 809-820). We also discuss some possible implications of sequence propagation inspired by mammalian literature, which we have now extended:

“Evidence from rodents has often associated spiking sequences with memory or spatial tasks (Buzsáki and Tingley, 2018; Diba and Buzsáki, 2007; Dragoi and Tonegawa, 2011; Modi et al., 2014; Vaz et al., 2020) or reported sequences within the spontaneous or stimulus-evoked activity in sensory areas (Carrillo-Reid et al., 2015; Dechery and MacLean, 2017; Fellous et al., 2004; Luczak et al., 2015; Luczak and MacLean, 2012). More generally, spiking sequences have been proposed as parts of an information-coding scheme for communication within cortex (Luczak et al., 2015) or as content-free structures that can reference experiences distributed across multiple cortical modules (Buzsáki and Tingley, 2018). Although spiking sequences were observed in the ex vivo turtle cortex, we predict the existence of sequences under in vivo levels of spontaneous activity. We propose that reliable routing of sequential activity might implement the visual tuning of cortical neurons.

(…)

The presence of combination-specific followers in our model suggests that the tuning of a neuron may be expressed as the result of logical operations on the tuning properties of other neurons in the recurrent circuit.”

20. The gating results seem very exciting! It might be nice to highlight connections to a recent explosion in ML papers on gating and putting that part of the work more front and center in the abstract.

We thank the reviewers for their appreciation. We have added the following text to our discussion:

“We expect that complex computations can be built from this proposed form of gating. Previous theoretical work on signal propagation in recurrent networks with irregular background activity found that the combination of parallel pathways and gating can implement different logic statements, illustrating how gating can form the basis for more complex computations (Vogels and Abbott, 2005). Indeed, recent advances in machine learning result from leveraging gating to control the flow of information more explicitly. For instance, gating can implement robust working memory in recurrent networks (Chung et al., 2014), prevent catastrophic forgetting through context-dependent learning (Masse et al., 2018), and produce powerful attention mechanisms (Vaswani et al., 2017).”

21. Line 183: "the number of excitatory followers far exceeds the number of inhibitory followers". This seems obvious in a network with 93% excitatory neurons. Are we missing something?

Our model is based on experimental sequences that mostly consist of inhibitory followers (Hemberger et al., 2019). This was likely due to the experimental setup, where an MEA was placed under the bottom inhibitory layer of the 3-layered turtle cortex, resulting in very few excitatory neurons being spike-sorted. However, the length of experimental sequences (up to 200 ms), suggested that propagation should be happening within the excitatory layer, even though it was not electrophysiologically accessible. Thus, one of the motivations for our computational study was to verify that excitatory followers are common and to study propagation within the excitatory layer (Introduction, lines 54-56). We have replaced this sentence to make this more explicit:

“While the electrical distance between the MEA and the pyramidal cell layer limited the experimental observation of excitatory followers in the turtle cortex, our model predicts that their number far exceeds that of inhibitory followers”

22. Line 187: Hz is a unit for oscillations and spiking activity is not oscillatory. Might we suggest spikes/s?

We have changed all instances of Hz for “spk/s”, as in the experimental publication that our model is based on (Hemberger et al., 2019).

23. Figure 2E: The MEA square is an excellent example of the sort of thing that this reviewer cannot see when printed at the intended size. Please have mercy on older eyes.

We apologize for the lack of clarity. We have changed the MEA square color and line thickness to improve contrast.

We have also updated all of our main figures to ensure consistent maximum width (A4 minus margins: 159.2 mm) and font size (8pt), to improve whitespace use between panels, and reduce too light shades (see reviewers’ comments 28 and 30).

24. Line 250-254: We struggled a bit to understand the concept of spike transfers. It seems like this is any link in any sequence where the presynaptic neuron is excitatory. Could you expand this explanation, please?

Thanks for this comment. Indeed, spike transfer refers to any excitatory transmission within 100 ms. We studied spike transfers for the full network activity, not only on sequences. We have updated the text to read (lines 281-293):

“In a random subset of our 6,000 simulations (n=900), we searched for instances when a spike successfully traversed a connection, i.e., when the postsynaptic neuron fired within 100 ms of the presynaptic one. We call this successful activation of a connection a “spike transfer” from the pre- to the postsynaptic neuron. We thus combined spikes with recurrent connectivity to produce a directed acyclic graph of all spike transfers for each simulation (Figure 3A). Note that we studied spike transfers among all connected neurons in the network, allowing us to characterize the connections that are most frequently traversed. Most excitatory-to-excitatory spike transfers in low-activity simulations show a delay from pre- to postsynaptic spike of 6-8 ms (Figure 3B), matching the delays measured in turtle cortex (Hemberger et al., 2019). Interestingly, excitatory-to-inhibitory spike transfers display consistently shorter delays than their excitatory-to-excitatory counterparts, even at higher firing rates (Figure 3B inset), possibly reflecting the more depolarized state of inhibitory neurons (Figure 1I).

Please see also our response to reviewers’ comment 11.

25. Line 265: "very few motifs lead to the activation of excitatory neurons". This confused us because in Line 183 it says "the number of excitatory followers far exceeds the number of inhibitory followers". We are clearly missing something important; please clarify.

We apologize for the lack of clarity. The first sentence means that the spike transfers that lead to an excitatory spike rarely form complex motifs but are rather one-to-one transfers. This does not mean that there are very few excitatory spikes. We have updated that line to read:

“In low-activity simulations, we found that excitatory spikes are rarely triggered by convergence or fan motifs (Figure 3C top, conv.), but they are rather the result of one-to-one spike transfers (Figure 3C top, single).”

26. Line 305: How strong do the connections have to be? We're guessing strong enough so a single presynaptic spike causes a postsynaptic spike, but this should be stated.

We have rewritten this line to:

“Hence, we conclude that sparse connections strong enough to trigger a postsynaptic spike are necessary and sufficient to produce repeatable sequences in our model.”

We note that the exact strength of these connections will depend on the network state. From our main text (lines 116-119):

“The actual efficacy of these rare but powerful connections depends on the conductance state of their postsynaptic neurons and, thus, on ongoing network activity and the state of adaptation currents.”

27. Line 316-319: We feel that the role for strong connections is more clearly demonstrated than the role for weak connections. Obviously, if there are only a very few strong connections then the neurons will have a lower mean membrane potential and will be less responsive to spiking input. Is it that simple, or is there more to it?

Figure 3H directly compares simulations with and without weak connections (see the updated figure and caption in reviewers’ comment 28 below).

If there are only very few strong connections for some ranges of input (approx. 55-70 pA), the membrane potential is too far from threshold for most of these connections to be effective. In that case, the presence of weak connections can promote reliable transmission.

In other ranges of input, weak connections might increase the mean membrane potential to be too close to threshold, resulting in less predictable firing. In that case, the presence of weak connections reduces the reliability of the responses.

This effect is similar to stochastic resonance, which we describe in our Discussion section (lines 745-754) (Teramae et al., 2012), and to untuned depolarization, as reported in vivo in mouse V1 (828-830) (Cossell et al., 2015).

We also address the role of weak connections in our response to the reviewers’ comment 17.

28. Figure 3H: We lack the visual acuity to interpret this plot. Perhaps an alternative representation would be clearer?

We have simplified this plot and added explicit labels to the input regimes where the role of weak connections changes (see reviewers’ comments 17 and 27). We have also increased the size and updated its caption:

“H. Difference between the number of followers detected in full and strong-only models under the same input. Brackets indicate regimes where the presence of weak connections increases (full>str) or decreases (full<str) the follower count. 2,000 simulations.”

29. Line 366: two "due to"s in this sentence.

Thank you.

“Importantly, the few equally strong connections between sub-networks do not always guarantee the reliable propagation of activity between them due to unpredictable network effects resulting from recurrent interactions”

30. Figure 4E-G: there are numbers on these plots that we definitely can't read, and the connectivity in 4G is not only tiny but also a very pale grey in places.

We apologize for the lack of clarity. As we describe in comment 23, we have now updated all of our main figures to ensure consistent maximum width (A4 minus margins: 159.2 mm) and font size (8pt), to improve whitespace use between panels, and remove shades that are too light.

31. Figure 4E, F: this is a measured frequency rather than a probability, right?

Correct. These are measured frequencies, which we take as our best estimate of the corresponding probabilities. We have edited the captions to specify this:

“E. Relationship between connectivity and follower clusters. Left: schematic illustrating connections between (purple) and within (blue) clusters. Right: Estimated probability (measured frequency) of strong (top 0.3%) excitatory-to-excitatory connection within or between clusters for 10,472 clusters of at least 5 followers pooled across all 6,000 simulations.

F. Estimated probability (measured frequency) of postsynaptic activation conditioned on presynaptic activation in the same trial for excitatory-to-excitatory connections, pooled across all 6,000 simulations. Boxes: median and [25th, 75th] percentiles.”

32. Line 618: landmark -> hallmark.

Thank you.

33. Line 651: it seems to us that your sequence structures have got a lot in common with those reported in your ref 34 (Polychronous groups, Izhikevich), which also depends on very strong synapses. Can you expand this section with some comparison to that study?

We have added the following to our Discussion section (lines 710-714):

“A related structure to synfire chains is that of polychronous chains, where synchrony is replaced by precise time-locked patterns that account for the distribution of synaptic delays (Egger et al., 2020; Izhikevich, 2006). These time-locked patterns, or polychronous groups, self-reinforce through spike-timing-dependent plasticity, resulting in stronger synapses (Izhikevich, 2006).”

34. Line 731: "may not be linked to behavior". Well, it's ex vivo, so we guess certainly not linked to behavior. Perhaps you can rephrase to make your meaning clearer?

We have simplified this sentence and edited the ones around it as part of our response to the reviewers’ comment 20. We refer the reviewers to that response for the updated text.

35. Line 818: "variable number of connections". It would be nice to know the statistics of this and how it compares with biology

We now give the average number of connections in the caption of Figure 1A (see our response to reviewers’ comment 1). We provide an example of the distribution of weak and strong E-E connections in Figure 1E.

36. Line 822: Nest -> NEST. (And we appreciate you stating and citing the version used!)

Thank you, and apologies for the misspelling.